# Auxin-dependent control of a plasmodesmal regulator creates a negative feedback loop modulating lateral root emergence

Ross Sager[1], Xu Wang[1,6], Kristine Hill [2,7], Byung-Chun Yoo[3], Jeffery Caplan[1,4,5], Alex Nedo[5], Thu Tran[1], Malcolm J. Bennett [2] & Jung-Youn Lee [1,4,5]*

Lateral roots originate from initial cells deep within the main root and must emerge through several overlying layers. Lateral root emergence requires the outgrowth of the new primordium (LRP) to coincide with the timely separation of overlying root cells, a developmental program coordinated by the hormone auxin. Here, we report that in *Arabidopsis thaliana* roots, auxin controls the spatiotemporal expression of the plasmodesmal regulator PDLP5 in cells overlying LRP, creating a negative feedback loop. PDLP5, which functions to restrict the cell-to-cell movement of signals via plasmodesmata, is induced by auxin in cells overlying LRP in a progressive manner. PDLP5 localizes to plasmodesmata in these cells and negatively impacts organ emergence as well as overall root branching. We present a model, incorporating the spatiotemporal expression of PDLP5 in LRP-overlying cells into known auxin-regulated LRP-overlying cell separation pathways, and speculate how PDLP5 may function to negatively regulate the lateral root emergence process.

---

[1] Department of Plant and Soil Sciences, University of Delaware, Newark, DE 19711, USA. [2] Centre for Plant Integrative Biology, University of Nottingham, Nottingham LE12 5RD, UK. [3] Gene Editing Institute, Christina Health Care System, Newark, DE 19711, USA. [4] Delaware Biotechnology Institute, University of Delaware, Newark, DE 19711, USA. [5] Department of Biological Sciences, University of Delaware, Newark, DE 19711, USA. [6]Present address: Institut für Physiologie und Biotechnologie der Pflanzen, Universität Hohenheim, Emil-Wolff-Straße 23, 70593 Stuttgart, Germany. [7]Present address: Zentrum für Molekularbiologie der Pflanzen, Universität Tübingen, Auf der Morgenstelle 32, 72076 Tübingen, Germany. *email: lee@dbi.udel.edu

Lateral root branching is critical for rapidly increasing the surface area of the root system architecture to aid nutrient and water uptake[1]. Lateral root primordium (LRP) originate from xylem pole pericycle derived "founder cells" which, through a series of formative cell divisions, creates a growing dome-shaped primordia that eventually passes through endodermal, cortical, and epidermal layers of the primary root[2,3]. This developmental program requires intercellular coordination between emerging LRP and overlying cells, facilitated by auxin released from the tip of growing LRP to trigger cell wall separation[2–5]. In *Arabidopsis thaliana*, specific auxin influx and efflux carriers LAX3 and PIN3 regulate the flow of auxin from LRP into overlying cells in a highly localized manner. This ensures that only cells in direct contact with the new organ separate and allows LRP to emerge while maintaining the integrity of surrounding root tissues[6]. To date, researchers have focused their attention on auxin transport to understand the mechanisms underlying the LRP emergence and cell separation program[4,6–9].

In addition to being transported via specialized membrane localized carriers, auxin can move freely between plant cells through cytoplasmic pores termed plasmodesmata when they are open[10,11]. Plasmodesmal permeability involves reversible accumulation and degradation of a callose "plug" around plasmodesmata[12,13]. During auxin-dependent LRP emergence, temporary symplastic isolation of new primordia via reversible callose accumulation is a critical factor determining organ formation and density[14]. Although not yet known whether changes in plasmodesmal callose levels in LRPs are linked to auxin movement, this study suggests that plasmodesmata could potentially influence this auxin-dependent process.

We have previously characterized the receptor-like transmembrane protein PLASMODESMATA-LOCATED PROTEIN (PDLP) 5 in aerial tissues, where it exclusively localizes to plasmodesmata and restricts cell-to-cell movement via stimulating plasmodesmal callose deposition[15–17]. Plasmodesmal callose levels are reduced while cell-to-cell movement is accelerated in the *pdlp5-1* knock-down mutant compared to wild type (WT) plants. In contrast, plasmodesmal callose levels are increased and plasmodesmal trafficking is severely suppressed in PDLP5-overexpressing plants (*PDLP5OE*)[15].

In the current study, we report that auxin-dependent PDLP5 expression stimulates the formation of a temporary symplasmic domain in LRP-overlying cells, ensuring optimal levels of auxin are reached so that organ emergence occurs in a synchronized manner. We show that auxin induces PDLP5 expression specifically in LRP-overlying cells in a highly localized spatiotemporal manner during organ emergence, and that accumulation of auxin and auxin-dependent genes in these cells are altered in *pdlp5-1* and *PDLP5OE* plants.

## Results

**Auxin induces PDLP5 expression in LRP-overlying cells.** To determine which root cell types express *PDLP5*, roots of a transgenic *PDLP5pro:GUS* reporter line were analyzed (Fig. 1a). We compared the *PDLP5pro:GUS* pattern to the auxin response reporters *DR5:GUS* and *LAX3pro:GUS* under the same experimental conditions. While the GUS reporter was detected in lateral root forming zones and within LRPs in *DR5:GUS* roots, in *PDLP5pro:GUS* roots it was excluded from LRPs but induced in LRP-overlying cells (Fig. 1a; Supplementary Fig. 1). In *PDLP5pro: GUS* GUS roots, expression was also detected in the protoxylem and groups of cells along the main root axis (Supplementary Fig. 1). Furthermore, *PDLP5pro:GUS* expression occurred in a distinct spatiotemporal manner during all eight stages of LRP development[2], starting in endodermal (En) cells during organ initiation and early development, then subsequently in cortical (Co) and epidermal (Epi) cells as primordia emerged through these cell layers (Fig. 1a).

The *PDLP5pro:GUS* staining pattern in LRP-overlying cells was similar to several known auxin-regulated genes involved in LRP emergence such as *LAX3* except that the latter gene is not expressed in En cells (Fig. 1a). It is known that *LAX3pro:GUS* expression in LRP-overlying cells is driven by shoot-supplied auxin[4], which prompted us to examine if *PDLP5* expression is similarly regulated by shoot-supplied auxin. For this, shoots were removed from the three *GUS* reporter lines two days before their remaining roots were stained. The result revealed a substantial reduction of GUS staining in *PDLP5pro:GUS* roots (Fig. 1b) as well as in *LAX3pro:GUS* and *DR5:GUS* control roots (Supplementary Fig. 2). We concluded that shoot-derived auxin likely controls the highly-localized *PDLP5* expression in LRP-overlying cells. Next, we tested auxin-dependent induction of *PDLP5* expression using the GUS reporter in the presence and absence of auxin transport inhibitor 1-N-naphthylphthalamic acid (NPA) or the synthetic auxin analog 1-naphthaleneacetic acid (NAA). Five micrometer NPA treatment abolished GUS expression in both *DR5:GUS* and *PDLP5pro:GUS* roots (Fig. 1c) while 1 μM NAA induced intense GUS staining (Supplementary Fig. 3a). A lower concentration of NAA (0.1 μM) allowed us to discern that NAA induced GUS staining specifically and distinctively in regions where LRPs were formed (Fig. 1d; Supplementary Fig. 3b).

We also examined if other hormones such as SA, which induces *PDLP5* expression in leaves[16] has a similar effect inducing *PDLP5* in roots. This experiment showed that exogenous SA treatment could induce a strong *PDLP5* expression in roots as determined by GUS staining and RT-PCR (Supplementary Fig. 3a, c). However, in contrast to discrete GUS stains induced by auxin along the roots, SA-treated roots showed a uniform staining pattern. Other hormones such as cytokinin, jasmonic acid, and abscisic acid, had little to no effects on *PDLP5* induction in roots, although cytokinin seems to reduce the area of *PDLP5pro:GUS* expression within the protoxylem (Supplementary Fig. 3a). Lastly, auxin treatment increased *PDLP5* transcript levels in WT seedling roots (Supplementary Fig. 3c) and PDLP5-GFP accumulation at plasmodesmata between the epidermis and cortex cells in *PDLP5pro:PDLP5-GFP/pdlp5-1* roots (Fig. 1e). Collectively, these data corroborate that *PDLP5* expression is an auxin-dependent response.

Next, to gain insights into the regulatory mechanism controlling *PDLP5* expression in roots, we examined the impact of the auxin response mutants, *iaa28-1* and *shy2-2* on *PDLP5pro:GUS*. *IAA28* and *SHY2* genes encode repressors that inhibit the transcription factors ARF7 and ARF19, respectively, during LR formation[18,19]. LRP development is suppressed in *iaa28-1*[20] while ectopic LRP initiation is increased in *shy2-2*[21]. GUS staining revealed *PDLP5* expression was barely detectable in cells above a few early-stage LRP formed in *iaa28-1* roots (Fig. 1f; Supplementary Fig. 4). In contrast, *PDLP5* expression was strongly upregulated in En cells above the large number of unemerged LRP in *shy2-2* roots (Fig. 1f; Supplementary Fig. 4). We examined *shy2-2* roots under the microscope and could find few to no LRP past stage V, as reported elsewhere[21]. The differential regulatory effect of *iaa28-1* and *shy2-2* may reflect the recent report[22] that SHY2/IAA3 requires ARF targets to be SUMOylated before it can interact and repress their transcriptional activity. Next, we performed chromatin immunoprecipitation (ChIP) assays using anti-ARF19 which revealed binding to *PDLP5* promoter fragments (Fig. 1g). Collectively, these results indicate that the *PDLP5* expression in overlying cells during LR development requires auxin through the regulatory molecules IAA28 and ARF19, but not likely through SHY2.

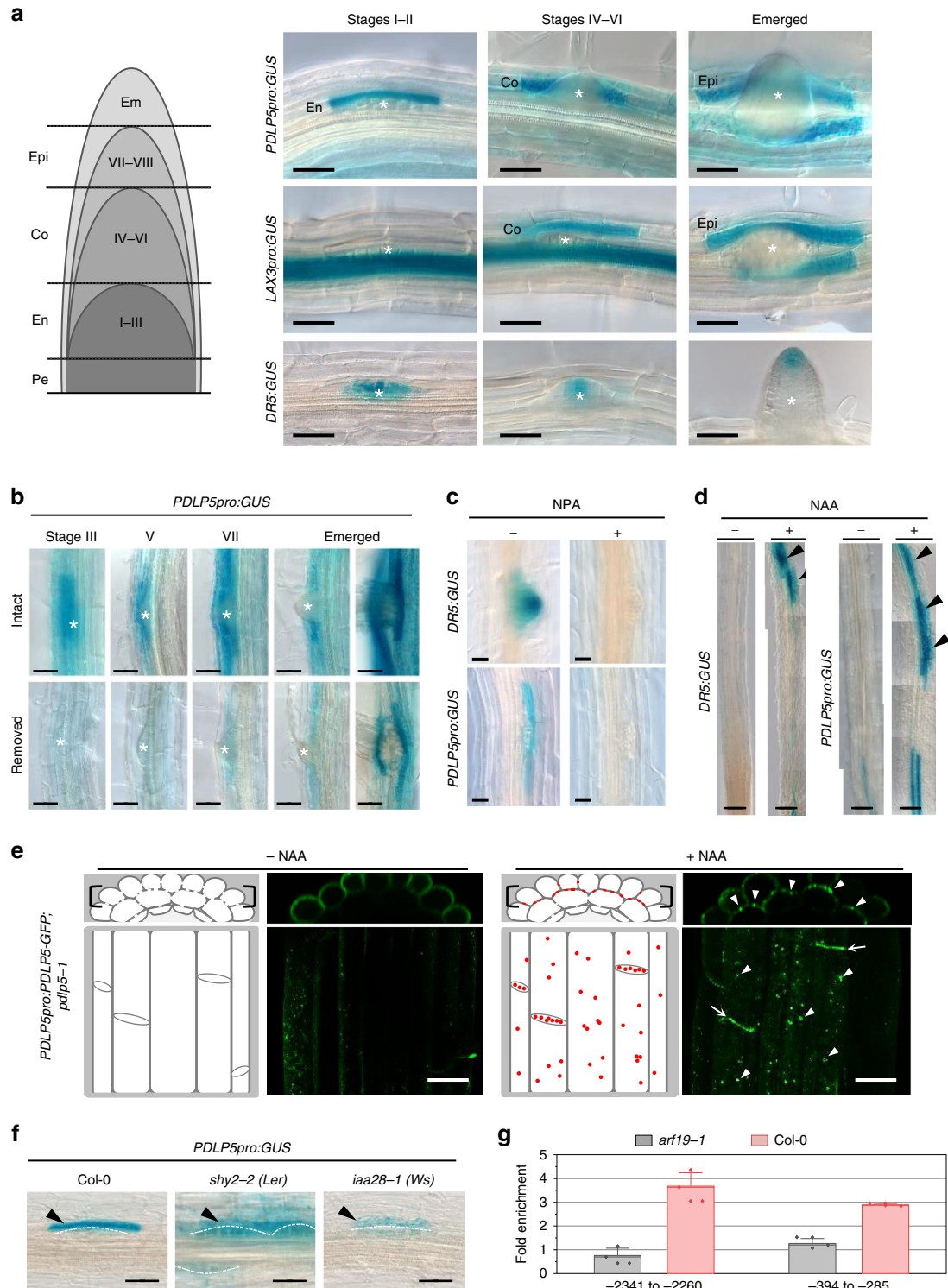

**PDLP5 Localizes to plasmodesmata in LRP-overlying Cells.**
Following observing that PDLP5-GFP localizes to plasmodesmata in root cells treated with auxin (Fig. 1e), we investigated if the protein preferentially accumulates at plasmodesmata of LRP-overlying cells during LR emergence. To this end, we tracked PDLP5-GFP localization to plasmodesmata in LRP-overlying cells at different emergence stages (Fig. 2a). *PDLP5pro:PDLP5-GFP* exhibited highly specific spatiotemporal expression patterns in LRP-overlying cells, appearing first in En cells during early LRP development followed by overlying Co and then Epi cells as

new primordia grew outwards (Fig. 2a). In each stage, PDLP5-GFP fluorescence revealed a typical plasmodesmal localization pattern—punctate signals at the cell wall junctions: at stage II, PDLP5-GFP signals were associated with two En cells positioned directly above the newly-forming LRP and specifically plasmodesmata at En–En, En–Pe, and En–Co cell wall junctions. At stages III and V, plasmodesmal signals were detected primarily in the walls surrounding two Co cells located directly above the two initial En cells (Fig. 2a, darts). Strong plasmodesmal labelings were detected at the Co–En and Co–Epi junctions as well as at the

**Fig. 1 Auxin regulates spatiotemporal PDLP5 expression in LRP-overlying cells. a** Representative GUS-staining images in LRP and overlying tissues during pre-emergence (I–II), emerging (IV–VI), and post-emergence stages in *PDLP5pro:GUS*, *LAX3pro:GUS*, and *DR5:GUS*, with a model of the emergence stages for reference. Pe, pericycle; En, endodermis; Co, cortex; Epi, epidermis. Scale bars, 50 μm. **b** *PDLP5pro:GUS* staining 2 days post-shoot removal, at various stages of LRP emergence. Scale bars, 50 μm. Asterisks indicate the center of LRP tip. **c** GUS-stained seedlings of *PDLP5pro:GUS* and *DR5:GUS*, transferred at 5 dpg to media lacking (−) or containing (+) 5 μM NPA, and grown for a further 24 h. Scale bars, 25 μm. **d** 0.1 μM 1-NAA treatment of *PDLP5pro:GUS* and *DR5:GUS* roots. Scale bars, 0.5 mm. **e** Induction of PDLP5-GFP in 10 μM NAA-treated *PDLP5pro:PDLP5-GFP;pdlp5-1* roots. Mock-treated epidermal cells exhibit non-specific background green fluorescence within and at the outer surface of the cells exposed to the media. In contrast, NAA treatment induces PDLP5-GFP expression, which accumulate as punctate fluorescent signals at plasmodesmata in cross walls between Co cells (arrows) and tangential walls between Co and Epi junctions (darts). Cartoons show absence and presence of PDLP5-GFP signals at plasmodesmata (red dots) in root cross-sectional and longitudinal representations of confocal images. Scale bars, 25 μm. **f** Close-up of GUS-stained LR initiation sites showing expression of *PDLP5pro:GUS* in Col-0, *shy2-2* (*Ler*), and *iaa28-1* (Ws) backgrounds. Scale bars, 50 μm. LRP are makred by dashed arcs. **g** ChIP assay showing the upstream regions of PDLP5 (−2341 to −2260 where +1 is the start codon) and (−394 to −285) amplified by ChIP primers (see Supplementary Fig. 5). Fold enrichment is calculated as the amount of promoter fragment immunoprecipitated relative to the non-immunoprecipitated input chromatin. Results are representative of three biological repeats. Bars, standard error.

cell wall junctions between those two Co cells. Notably, intense but non-punctate fluorescent signals were sometimes detected at En–En and Co–Co cell junctions that would soon separate (Fig. 2a, carets). While PDLP5-GFP signals disappeared in separated walls (Fig. 2a, double arrows), PDLP5-GFP signals at the plasmodesmata of the other junctional walls persisted after LR emergence (Fig. 2a, b). Finally, PDLP5-GFP expression and localization patterns described in LRP-overlying cells were identical in the *pdlp5-1* background complemented with the functional *PDLP5pro:PDLP5-GFP* reporter (Supplementary Fig. 6a).

**PDLP5 restricts cell-to-cell movement in roots.** Based on its demonstrated role as a plasmodesmal regulator in aerial tissues, we reasoned that PDLP5 may function to restrict cell-to-cell movement in LRP-overlying cells during LR emergence. Since PDLP5 restricts plasmodesmal permeability by stimulating callose deposition in leaf cells, we hypothesized that it might function similarly in root cells. We evaluated plasmodesmal callose levels in LRP-overlying cells using aniline blue staining. Aniline blue binds to callose to give a yellow fluorescence in ultraviolet light[23]. Although staining was successful in detecting plasmodesmata (Supplementary Fig. 6b), it was not possible to stain plasmodesmal callose consistently to discern if there were measurable differences in plasmodesmal callose levels between WT and *pdlp5-1* roots. Therefore, we utilized the *Arabidopsis* transgenic line *pER8:PDLP5* that we had described elsewhere[14], which expresses *PDLP5* under the control of an estradiol-inducible promoter, to assess if we could correlate ectopic PDLP5 induction and plasmodesmal callose levels in root tip cells where callose staining was possible. This experiment revealed that estradiol treatment increased a statistically significant amount of callose deposition at cell-cell junctions in *pER8:PDLP5* roots compared to mock-treated roots (Fig. 2c; Supplementary Fig. 7). These results suggest PDLP5 functions in root cells to restrict cell-to-cell movement via stimulating callose deposition.

Since directly assessing plasmodesmal permeability across a few internal root cells is not currently possible, we designed a new experimental set-up that would allow us to evaluate PDLP5 function indirectly in restricting cell-to-cell movement across the En cell layer. To this end, we created a movement reporter line, in which expression of free GFP is driven by an En-specific promoter derived from the genomic DNA encoding *Casparian membrane protein 1* (*CASP1*)[24]—*CASP1pro:GFP*. For a control reporting *CASP1* expression domain, we also created a non-mobile reporter replacing free GFP using ER-targeted citrine YFP—*CASP1pro:ER-YFP*. These reporter lines were then introduced into the estradiol-inducible *pER8:PDLP5* plants[15] to monitor alterations in GFP movement out of the En cell layer in the presence of ectopically induced PDLP5 in roots. In *CASP1pro:ER-YFP* roots, fluorescent

signals were confined to En layer as expected, and this pattern did not change by estradiol-treatment (Fig. 2d). In contrast, in *CASP1pro:GFP/pER8:PDLP5* F1 roots, GFP fluorescence was detected not only in the En layer but also in the neighboring cell layers, Co and Pe, indicating that plasmodesmata at the En–Co and En–Pe junctions are permeable to GFP under normal growth conditions. This movement pattern, however, was altered by PDLP5 induction; GFP could still move and accumulate in the Pe layer, but it could not move into the Co layer of 64% of the seedlings examined (Fig. 2d). This result suggests that PDLP5 is capable of restricting plasmodesmal permeability in root cells.

**PDLP5 is required for lateral root branching and emergence.** Next, we investigated if the spatiotemporal expression of PDLP5 in LRP-overlying cells has roles in LR development and root branching. To facilitate LR phenotyping using histochemical staining, we introduced *DR5:GUS* into *PDLP5OE* and *pdlp5-1* mutant backgrounds. Compared to *DR5:GUS* Col-0 controls 8-days post germination (dpg), approximately 30 and 70% fewer secondary and tertiary roots were formed in *PDLP5OE;DR5:GUS* seedlings, and by 11 dpg, 25 and 50% fewer secondary and tertiary roots formed, respectively (Fig. 3a, b; Supplementary Fig. 8, 9a, b). In contrast, 50 and 70% more tertiary roots were formed at 8 dpg and 11 dpg, respectively, in *pdlp5-1;DR5:GUS* seedlings (Fig. 3a, b). Nevertheless, in spite of increased tertiary root numbers, root density remained comparable to that in WT seedlings because the secondary root length in *pdlp5-1* mutants was also increased by 30% (Supplementary Fig. 9c, d).

To gain better insight into how *PDLP5* may impact the LR number and growth, we next evaluated LRP emergence rates using an LR-induction assay[25] to monitor and compare all stages of LRP development. Four-day-old WT, *pdlp5-1*, and *PDLP5OE* seedlings grown vertically on agar plates were rotated 90° to induce the formation of new LRP at the root bend as each seedling turns towards the new gravitropic vector (Fig. 3c). Developmental stages of the induced LRP in each seedling (*y*-axis) was recorded from 12 to 48 h post-induction (hpi) (*x*-axis), then the distribution of LRPs in each stage at each time point was calculated as a percentage (*z*-axis) (Fig. 3d). During LRP initiation and early developmental stages 0–IV observed at 12, 18, and 24 hpi, LRP stage distributions exhibited no significant differences among all three genotypes. However, during LRP emergence stages V–VIII observed at 36, 42, and 48 hpi, LRP emergence occurred faster in *pdlp5-1* seedlings, while it was severely delayed in *PDLP5OE*. Specifically, compared to WT, 32% more *pdlp5-1* LRP were in stage VIII at 42 hpi, and 17% more had emerged by 48 hpi; meanwhile, no *PDLP5OE* LRP had emerged even by 48 hpi (Fig. 3d). Normal LRP emergence rate was restored by crossing *pdlp5-1* with *PDLP5pro:PDLP5-GFP*,

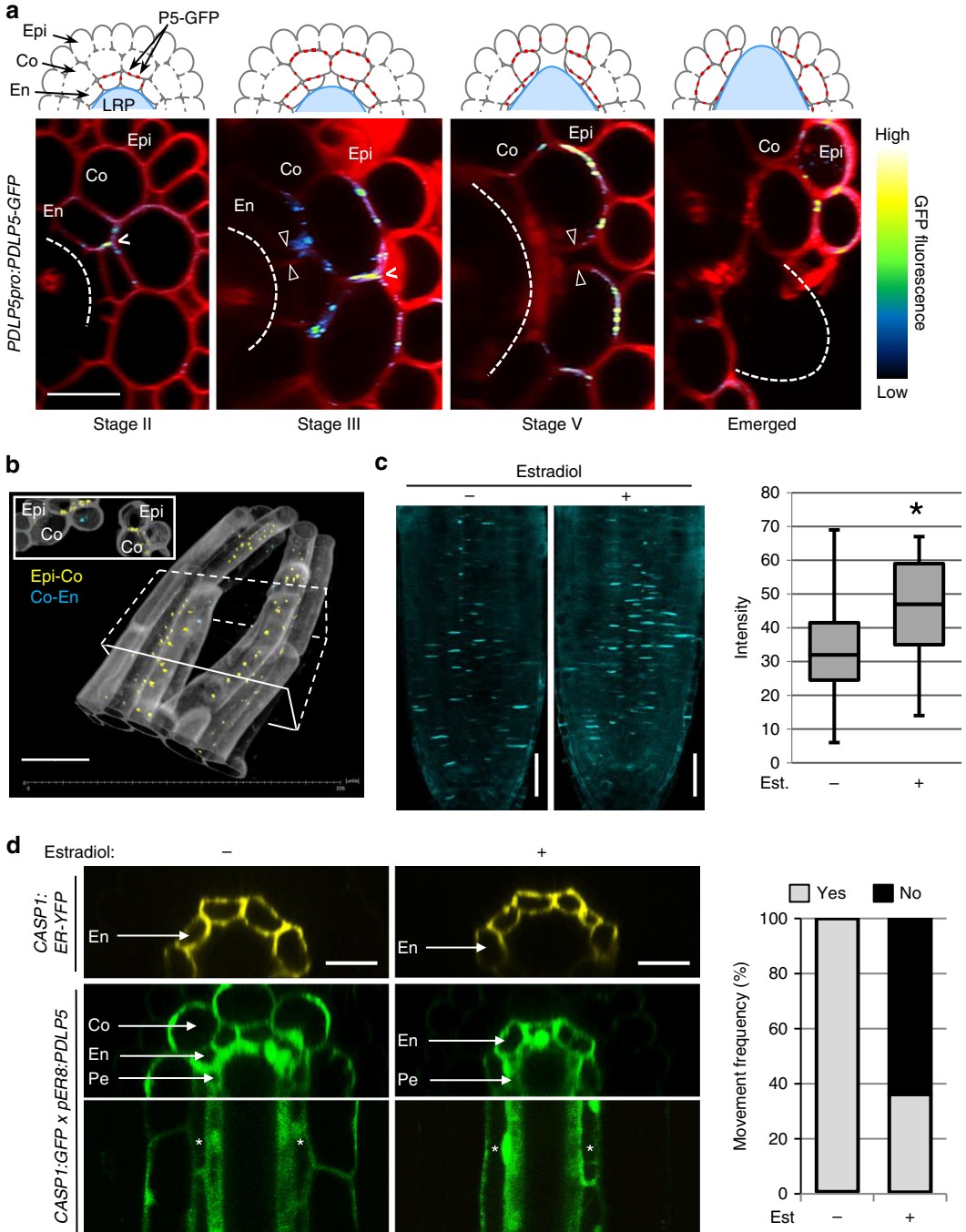

**Fig. 2 PDLP5 localizes to plasmodesmata in LRP-overlying cells. a** Representative confocal images showing 2D maximum intensity projections of 10 μm-thick cross-sections of *PDLP5pro:PDLP5-GFP*. Plasmodesmata marked by punctate PDLP5-GFP signal (in cartoons, red dots) can be seen at the cell junctions. PDLP5-GFP signals vary in color ranging from blue to green, yellow, and white depending on fluorescence intensity as indicated in calibration bar of the fluorescence intensity included on the right. The dashed arcs are positions of LRP; open arrowheads, separated cell walls; carets, strong PDLP5-GFP signal in cell walls just prior to separation. Scale bars, 25 μm. **b** A representative 3-D rendering of a confocal image, showing a 2-D maximum intensity projection of a 45 μm-thick cross-section of cells overlying an emerged LRP (not rendered) in *PDLP5pro:PDLP5-GFP* root. Signals at other junctions (Co–Co, Ep–Ep) were rare at the emergence stage we used for modeling. Inset, a cross-sectional view of the image taken from the boxed area. PDLP5-GFP-labeled plasmodesmata in different cell junctions are color-coded. **c** *pER8:PDLP5* root tips stained with aniline blue after 24 h of 10 μM estradiol or mock treatment. Estradiol- and mock-treated *n* = 25; scale bars, 25 μm. **d** PDLP5 induction can prevent GFP movement from En to Co root tissue in *pER8:PDLP5 × CASP1:GFP* F2 crosses. Mock *n* = 24, Treated *n* = 47; scale bars, 20 μm.

demonstrating that the LRP emergence defect in the mutant is due to loss of PDLP5 (Supplementary Fig. 9e). Collectively, our results suggest that PDLP5 negatively regulates the rate of LRP emergence.

**PDLP5 modulates auxin accumulation in LRP-overlying cells.** Compared to WT, *pdlp5-1* often had expanded *DR5:GUS* staining in the LRP zone and root tip, while *DR5:GUS* expression was generally weaker throughout *PDLP5OE* roots (Supplementary

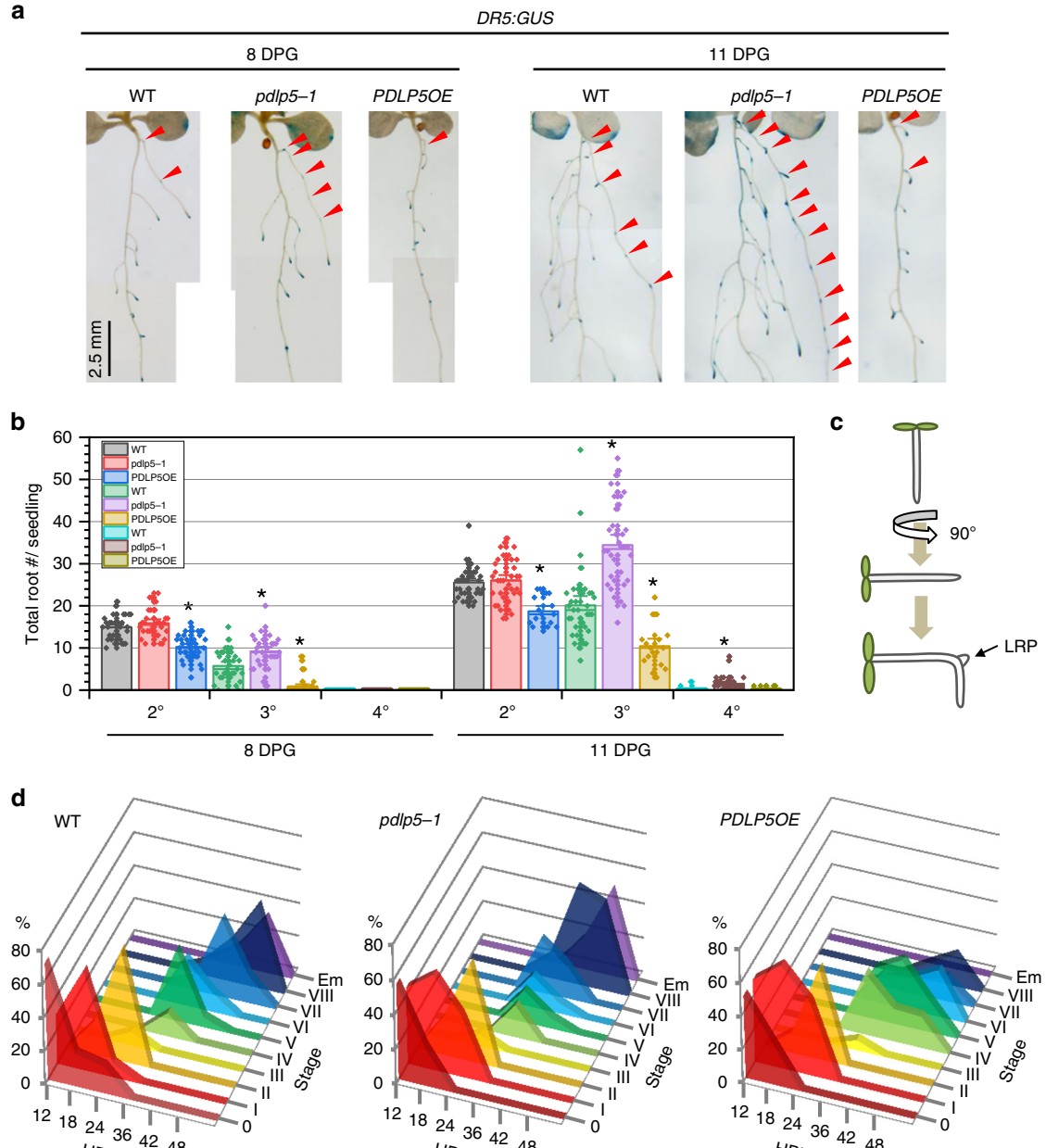

**Fig. 3 PDLP5 is required for normal LR emergence. a** LR development in WT, *pdlp5-1*, and *PDLP5OE* expressing *DR5:GUS*. Arrowheads, tertiary roots. **b** Quantification of total lateral root numbers (both emerged and un-emerged secondary [2°], tertiary [3°], and quaternary [4°] roots). $n \geq 30$ per seedling set. Bars, standard deviation. Asterisks, significance determined by student *T*-test ($P < 0.01$). **c** A diagram depicting the gravitropic assay and position of the root bend LRP. **d** Time-course of LRP development and emergence at the root bend of 4-day-old seedlings following gravistimulation. *x*-axis, hrs post-induction (hpi); *y*-axis, LRP developmental stage; *z*-axis, distribution of LRP at different stages per time point; $n \geq 20$ seedlings per set.

Fig. 10). This led us to consider whether *PDLP5* may modulate LR progression during emergence by affecting auxin accumulation and/or distribution within the newly forming LRP zone. To test this hypothesis using live cell imaging, we crossed the auxin reporter *DR5:3VENUS* with *pdlp5-1* and *PDLP5OE* plants. Fluorescence associated with 3VENUS in the nuclei allowed us to monitor auxin distribution from induced LRP into the overlying cells in real time (Supplementary Fig. 11a). Those emerging LRPs were observed between 22–36 hpi, and 27 hpi was found to be an optimal time point to quantify nuclei under our experimental conditions for two reasons: at this time point, the stage IV–V LRP would be approaching the Co cells, and the LRP and overlying Co nuclei were more distinguishable from each other. The number of fluorescent overlying Co cells increased in *pdlp5-1* roots, while it

decreased in *PDLP5OE* (Fig. 4a; Supplementary Fig. 11b). We have also examined the DR5-3VENUS signal intensities in LRP-overlying Co cells, but the variance in the fluorescence intensity of DR5:3VENUS was too high to show any statistical differences (Supplementary Fig. 13). The Box plot analysis of overlying Co cell numbers revealed that while 50% of WT seedlings had 3–5 *DR5:3VENUS*-positive Co cells, this range was skewed lower in *PDLP5OE*, with 50% of seedlings having only 2–4 fluorescent Co cells, and skewed higher in *pdlp5-1* mutants, with 50% having 4–5 positive Co cells (Fig. 4b). These results suggest that *PDLP5* negatively regulates the spread of auxin through overlying Co cells during LRP emergence.

To gain insight into the functional relationship between PDLP5 and downstream auxin-responsive genes required for

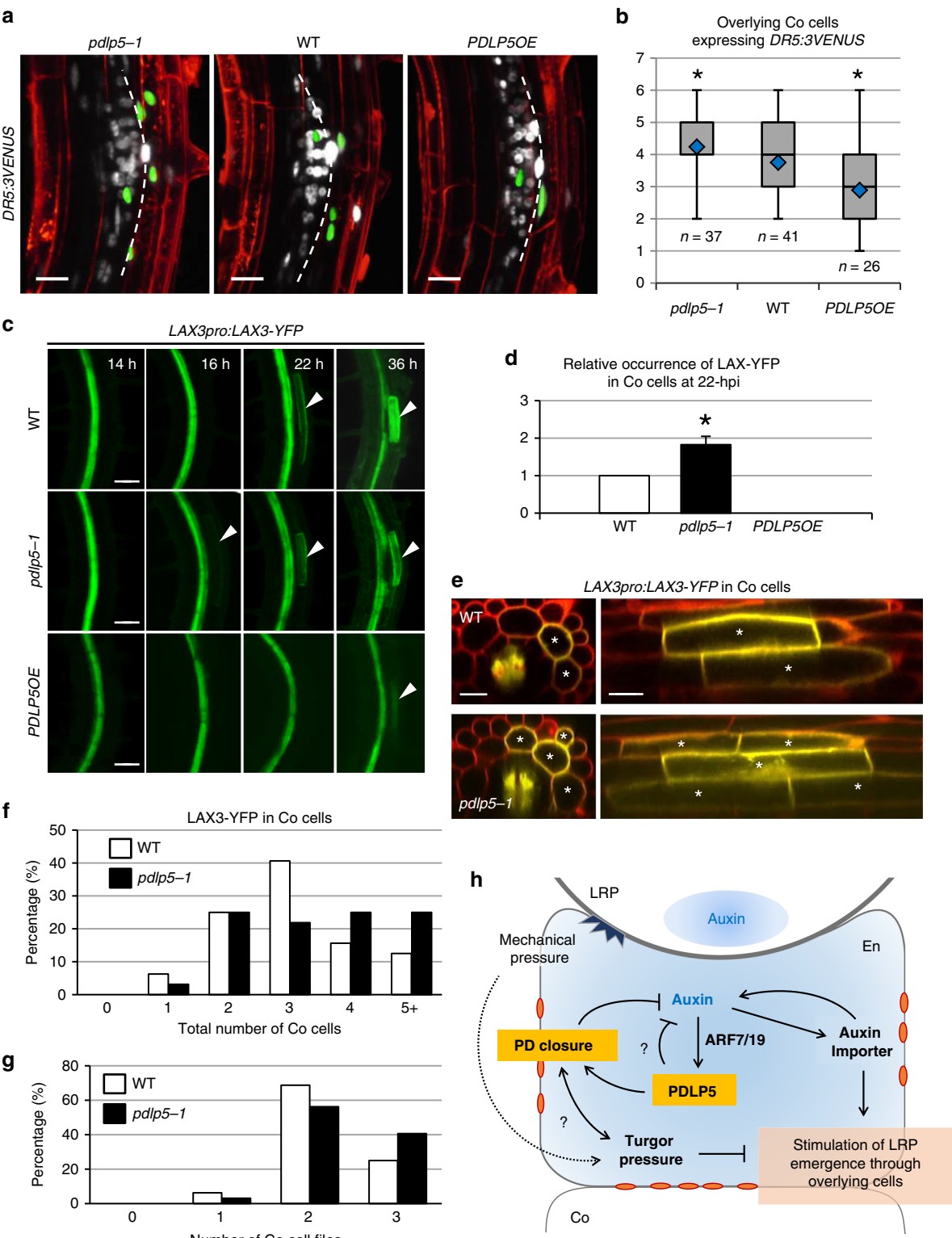

LR emergence, we investigated whether the expression of *LAX3*, a key late auxin response gene induced in LRP-overlying root cells, might be affected by *pdlp5-1* or *PDLP5OE*. To this end, the expression of *LAX3pro:LAX3-YFP* was monitored in WT, *pdlp5-1*, and *PDLP5OE* LRP over the time course of 14–36 hpg. LAX3-YFP fluorescence was detectable in LRP-overlying Co cells as early as 16 hpi in *pdlp5-1*, at which time no LAX3-YFP signals could be detected in WT Co cells (Fig. 4c). By 22 hpi, almost two-fold more *pdlp5-1* mutant seedlings than WT

expressed LAX3-YFP in Co cells, whereas *PDLP5OE* seedlings were substantially delayed expressing LAX3-YFP until 36 hpi (Fig. 4c, d; Supplementary Table 1, 2).

To determine the impact of *PDLP5* loss on the location of *LAX3* expression, we quantified how many LRP-overlying Co cells expressed *LAX3pro:LAX3-YFP* in WT and *pdlp5-1* backgrounds between 26–36 hpi (stages IV–VI), (Fig. 4e). We found that LAX3-YFP was expressed in more LRP-overlying Co cells in *pdlp5-1* during this time range; ~50% of

**Fig. 4 PDLP5 affects *LAX3pro:LAX3-YFP* or *DR5:3VENUS* expression domain. a** Representative confocal images of *DR5:3VENUS* in each genetic background. Overlying Co cells were imaged as confocal *z*-stacks (2D maximum intensity projections of 115.4 μm thick confocal volumes), and the number of overlying Co cells with auxin-induced DR5:3VENUS was quantified at 27 hpg. The nuclei within LRP and Co cells are pseudo-colored white and green, respectively, to aid their 3D positioning in 2D images. Scale bars, 20 μm. **b** A box plot showing quantification of the number of LRP-overlying Co cells with *DR5:3VENUS* signal at 27 hpg. Asterisks, statistical significance determined by student *T*-test (*P* < 0.05) on three biological repeats. **c** Representative confocal images of *LAX3pro:LAX3-YFP* in a time course following gravistimulation. Arrowheads, LAX3-YFP expression in overlying Co cells. Scale bars, 50 μm. **d** Quantification of relative occurrence of LAX3-YFP signal in Co cells at 22 hpg, based on the data presented in Table S1. Asterisk, statistical significance determined by student *T*-test (*P* < 0.01). Bars, standard deviation. *n* ≥ 30 per seedling line. **e** Representative images of *LAX3pro:LAX3-YFP* showing signal in LRP-overlying cortical cells in either the WT or *pdlp5-1* background. Images are 2D maximum intensity projections of 3.69 μm thick confocal volumes. Scale bars, 25 μm. **f**, **g** Graphs showing the percentage of total seedlings with specific numbers of LRP-overlying Co cells (**f**) or cell files (**g**) expressing *LAX3pro:LAX3-YFP* in WT or *pdlp5-1*. **f**–**g** *n* = 32. **h** A model illustrating a negative feedback loop between auxin and PDLP5-dependent plasmodesmal closure integrated with positive and negative signaling pathways dependent on auxin and turgor pressure. Depicted is stage III–IV LRP, approaching endodermis. Orange ovals, closed plasmodesmata.

*LAX3pro:LAX3-YFP/pdlp5-1* seedlings had 4–5 Co cells with LAX3-YFP, compared to only ~25% of *LAX3pro:LAX3-YFP/WT* (Fig. 4f). Furthermore, ~40% of *LAX3pro:LAX3-YFP/pdlp5-1* seedlings had signal in 3 cell files, compared to ~25% of *LAX3pro:LAX3-YFP/WT* (Fig. 4g). Finally, LAX3-YFP appeared earlier in *pdlp5-1* LRP-overlying Epi cells compared to WT (Supplementary Fig. 12). Since *LAX3* expression is dependent on targeted auxin accumulation in LRP-overlying cells, our results suggest that loss of *PDLP5* could increase auxin diffusion from these cells, thereby expanding the *LAX3* expression domain.

Collectively, our study reveals that auxin upregulates PDLP5 progressively in root cells overlying newly forming LRP; PDLP5 localizes to plasmodesmal and can regulate cell-to-cell movement in roots, as it does in leaves; and the absence of PDLP5 allows expansion of auxin distribution in LRP-overlying cells, enhancing LR emergence.

## Discussion

Our study reveals a negative feedback loop linking the plasmodesmal regulator PDLP5 with auxin in LRP-overlying cells. This regulatory circuit may function to symplasmically isolate LRP-overlying cells while ensuring organ emergence occurs at a normal rate. Starting soon after founder cell division begins, auxin induces spatiotemporal expression of PDLP5 in the LRP-overlying cells, where PDLP5 localizes to plasmodesmata. We have shown that inducible PDLP5 expression in roots stimulates callose deposition and closes plasmodesmata, inhibiting cell-to-cell movement of En-expressed GFP into outer Co cells. PDLP5 negatively feedback-regulates auxin in LRP-overlying cells, altering the timing and number of overlying cells expressing auxin marker DR5 and LAX3. Lastly, the lack of PDLP5 increases higher-order LRP development and emergence rate.

Due to the highly spatiotemporal nature of PDLP5 expression in the LRP-overlying cells, and our previous data proving that PDLP5 closes plasmodesmata, we hypothesized that the overlying cells might become symplasmically isolated during LRP emergence. However, it was reported by Benitez-Alsonso et al.[14] that GFP expressed under the phloem-specific SUC2 promoter was present in parental root cells including LRP-overlying endodermal cells during stages I-II. Since PDLP5 is expressed in cells overlying nascent LRP, observing GFP within these cells would at first seem to contradict our hypothesis on PDLP5's role for plasmodesmal regulation in those cells. However, GFP can remain quite stable within tissue for many hours after expression. Thus, it is possible that the GFP had already moved from the phloem into outer cell layers prior to PDLP5 expression during early LRP formation, and the signal Benitez-Alsonso et al. detected was actually residual fluorescence from GFP present in that tissue. Another technique for observing the potential effects of PDLP5 on cell-to-cell connectivity in LRP-overlying cells

would, therefore, be to observe it in real-time as shown in the report by Oparka et al.[26]. That report noted the unloading of phloem-loaded carboxy fluorescin (CF) dye into emerging LRP but no dye movement from cells within LRP to LRP-overlying cells. Our experiment showed that CF was unloaded into the cells of nascent (stage I–II) but no dye movement occurred from the cells of the LRP into cells overlying them, both in wild-type and the *pdlp5-1* roots (Supplementary Fig. 13). Our results seem to indicate that LRPs are not symplasmically connected with overlying cells, even from the earliest stages of development.

While these results meant we could not use CF phloem loading to directly test PDLP5-dependent isolation of LRP-overlying cells, it led to the unexpected insight that one of the earliest steps during LRP development is its isolation from outer root cells. This isolation may prevent the loss of important growth signals from the nascent LRP into the overlying tissues. Supporting this, it was recently discovered that a cuticle layer, functioning as a diffusion barrier, is deposited at the outermost cell wall of developing LRP at stage I-III[27]; our results make it tempting to speculate that the symplasmic isolation of LRP from outer cells may coincide with the cuticle deposition. It would be an interesting future investigation to detail when and how the plasmodesmal disconnection occurs between LRP and overlying cells. Meanwhile, direct examination of the PDLP5's impact on cell-to-cell connectivity of LRP-overlying cells would have to wait until a new technique is developed, allowing real-time analysis of a movement tracer out of those cells.

We summarize the PDLP5-auxin functional relationship in our model (Fig. 4h) and integrate key factors, such as the ARF7/19 module and cell turgor, which are vital for normal LRP development and emergence. The ARF7/19 pathway upregulates auxin transporters such as PIN3 in En and LAX3 in Co and Epi cells, which may in turn reinforce positive feedback to elevate auxin accumulation in LRP-overlying cells[8]. At the same time, turgor pressure rises within the overlying cell as it is compressed by the growing LRP, slowing the LRP emergence rate. Together, these positive and negative signaling components contribute to ensuring a safe passage for the new organ, while creating internal space for the developing organ. Although our illustration focuses on these events in the En layer, based on the PDLP5 expression pattern, this regulatory programming would likely repeat itself in overlying Co and Epi layers during later stages of LRP emergence.

In addition, auxin movement into LRP-overlying En cell induces the expression of PDLP5 via an ARF7/19 pathway, which stimulates plasmodesmal closure. Alternatively, high levels of auxin accumulating in the En could be sufficient to induce PDLP5 and thus restricting the number of cells undergoing a feed-forward increase in auxin response. In this scenario, auxin-dependent PDLP5 expression in outer cell layers might occur cell-autonomously without needing auxin to flow from the LRP to the

endodermis. As for the signal that feedbacks to auxin, we consider an indirect effect from blocking cell-to-cell movement by closing plasmodesmata or a direct effect exerted from PDLP5 or both. One of the indirect effects of plasmodesmal closure could be on maintaining the turgor of the LRP-overlying cell against the mechanical pressure imposed by the growing LRP. The maintenance of turgor in overlying cells has been shown to be necessary to prevent a rapid loss of the cell volume and to slow the emergence process[28], which is consistent with the emergence phenotype exhibited in pdlp5-1.

PDLP5 expression in LRP-overlying cells appears to be regulated by shoot-driven auxin and mediated by the regulatory module ARF7/19. PDLP5 appears to be an early auxin response gene based on its impact on the expression of the late auxin response gene LAX3, and our ChIP assay results revealing PDLP5 auxin-dependent induction is mediated via ARF7/19. This regulatory module is essential for LRP initiation and emergence processes and controls the spatiotemporal expression of key downstream effectors, such as SHY2 and LAX3[4,22]. Although all auxin-responsive genes depend on the ARF7/19 module for induction, SHY2 is expressed in the overlying En, and LAX3 in the overlying Co and Ep, whereas PDLP5 is expressed sequentially in all three layers. This differential expression pattern suggests additional determining factor(s) exist specific for SHY2 and LAX3, but not for PDLP5 expression.

In addition to providing insight into plasmodesmata–auxin interactions, our data raises an interesting question about how PDLP5-mediated plasmodesmal closure might help to elevate turgor pressure in LRP-overlying cells. Water can move between cells through plasmodesmata; hence, changes in plasmodesmal permeability would impact cellular turgor pressure. A developing LRP is thought to sense mechanical resistance in the overlying En cells and abort if turgor of these cells does not decrease as the growing LRP pushes against them[28]. Therefore, it is an interesting possibility that plasmodesmal closure in LRP-overlying cells may be necessary for modifying the turgor pressure needed for normal progression of LRP emergence. Equally, it would also be possible that turgor pressure (increased in overlying cells as LRP pushes through) may augment plasmodesmal closure. Indeed, changes in turgor pressure are known to alter plasmodesmal permeability[29].

Another interesting question our data raises is how auxin-dependent induction of PDLP5 during LR emergence might regulate root architecture, altering LR branching patterns. Both primary and secondary root lengths, as well as lateral root numbers, are reduced in PDLP5-OE seedlings, while secondary root length and tertiary root numbers are increased without affecting the primary root length in pdlp5-1 mutants. It is known that plant root architecture changes in response to internal and external nutrient states, and certain nutrients are linked to specific root morphological and architectural modifications[30]. For example, while severe nitrogen deficiency inhibits overall root growth, mild nitrogen deficiency stimulates lateral root emergence, and elongation[31]. Notably, this emergence and elongation is driven by auxin accumulating in later stage LRP, at a similar time to when the first differences between WT and pdlp5-1 root emergence rate can also be observed. Future investigation may uncover a yet unknown signaling mechanism that suppresses PDLP5 in the roots so that when such a nutrient foraging program is activated, greater lateral root emergence and elongation can occur.

Symplasmic isolation of overlying cells is likely facilitated by callose-dependent plasmodesmal closure, based on our data that PDLP5 ectopic expression in roots stimulates plasmodesmal callose deposition. We hypothesize a mechanism exists to fine-tune auxin accumulation controlling outward LRP growth and cell separation pathways. Symplasmic isolation in these two

domains may be necessary for the optimal build-up of turgor pressure in LRP and overlying cell, respectively.

In conclusion, our findings reveal a role for plasmodesmata fine tuning the LR emergence program via a negative feedback mechanism modulating auxin response in LRP-overlying cells.

## Methods

**Plant materials, growth conditions, and genetic crosses.** All *Arabidopsis thaliana* genotypes were in the Col-0 genetic background, except for *shy2-2* in *Ler*, and *iaa28-1* in Ws. Seedlings were grown vertically in 0.5× MS agar under continuous light at 22 °C. Plants in soil were grown in 16 h light at 22 °C. All the genetic crosses were genotyped to identify homozygous mutations when necessary (Supplementary Table 3). Genomic DNA was isolated from segregating F2 plants followed by PCR analyses using gene-specific primers.

**GUS assay and LRP quantification.** GUS solution (100 mM sodium phosphate buffer, pH 7.0, 10 mM EDTA, 0.5 mM each potassium ferrocyanide and potassium ferricyanide, 1.24 mM X-Gluc, and 0.1% Triton X-100) was vacuum-infiltrated into plant tissue for 5 min, then removed from vacuum and incubated in 37 °C for 3–12 h, followed by a series of ethanol washes. Stained tissues were imaged using a Zeiss Axioskop 2 microscope. LRP were quantified by counting both the emerged LR and unemerged LRP, as determined by *DR5:GUS* staining of the primordia, under a dissecting microscope (1.2× magnification). LRP stages were determined by examining ethanol-cleared, GUS-stained tissue using a 40× water lens.

**Chromatin immunoprecipitation, PCR, and qPCR analyses.** A ChIP assay was performed on Col-0 and a knock-out allele, *arf19-1*[32] using 2–3 g root tissue pre-treated with 1 μM NAA and fixed under vacuum with 1% formaldehyde for 15 min. Nuclei were extracted following the protocol described previously[33] and ChIP was performed, using home-made anti-ARF19 anitbody[33] following the method basically as described previously[34]. Briefly, 200 μl of sheared chromatin (average fragment size of 400 bp) was added to 1 ml Immunoprecipitation Buffer (50 mM Hepes, pH 7.5, 150 mM KCl, 5 mM MgCl$_2$, 0.1% Triton X-100) and incubated along with 3 μg of anti-ARF19 at 4 °C. Protein G Dynabeads® (Invitrogen) were then added and further incubated at 4 °C overnight. Input and ARF19 immuno-precipitated DNA was used for qPCR with SYBR green master mix and primers (Supplementary Table 4). Oligos were designed to two regions of the *PDLP5/HWI1* (At1g70690) promoter (Supplementary Fig. 5). These regions contain putative AREs likely corresponding to ARF19 binding sites. Anti-ARF19 immunoprecip-tated DNA is normalized to input chromatin using an internal control (TUB3) not bound by ARF19. All qPCR reactions were performed as triplicate technical replicates using a Light Cycler 480 qPCR machine and are representative of three biological repeats. Genomic-PCR and RT-PCR were performed as previous described[15], using primers listed in Supplementary Table 4.

**Confocal microscopy and image processing.** For *PDLP5pro:PDLP5-GFP* localization, seedlings were stained for 10–15 min in 5 μg/ml propidium iodide at 7 dpg. Fluorescent imaging was performed on a Zeiss AxioObserver Z1 inverted light microscope using a LSM 710 scanhead. For *DR5:3VENUS*, a LD LCI Plan-Apochromat 25×/0.8 Imm Korr DIC objective was used, with a 514 nm excitation laser and 515–550 nm (for VENUS) and a 585–758 nm (for propidium iodide) emission filters. For *PDLP5pro:PDLP5-GFP*, a C Apochromat 40×/1.20 W Korr objective was used, with a 488 nm excitation laser and 500-550 (GFP) emission filter, and detected with a BiGaAsP (Bi Gallium Arsenide Phosphide) Detector. Image brightness, contrast and gamma were adjusted to enhance the images via ZEN 2011 software. The 3D model of *PDLP5pro:PDLP5-GFP* in overlying cells was created using Amira 5.6 software to render separate channels, highlighting GFP signal and interpolating root cell shape from the propidium iodide outline. For counting *LAX3pro:LAX3-YFP* cells, a 25×/0.8 Imm Korr DIC objective was used, with a 514 nm excitation laser and a 575–610 nm BP filter (YFP) and a 543–735 nm BP filter (propidium iodide), and YFP was detected with the BiGaAsP Detector. For monitoring the timing of *LAX3pro:LAX3-YFP* expression, the cortical cell fluor-escence at the root bend was monitored at different time points using a Zeiss LSM 780 confocal upright light microscope using a W Plan-Apochromat 20×/1.0 DIC M27 75 mm objective and the 415-nm excitation line of an argon laser with 520–550 nm band pass emission filter. Images are presented as 3-D composites of 30 μm-thick z-stacks. Aniline-blue stained callose imaging was performed on a Zeiss LSM880 multiphoton confocal microscope, using an LD LCI Plan-Apochromat 25×/0.8 Imm Korr DIC M27 objective, with a 780 nm multiphoton excitation laser and 410–552 nm emission filters.

**Reporting summary.** Further information on research design is available in the Nature Research Reporting Summary linked to this article.

## Data availibility

Data supporting the findings of this study are available in the manuscript and its supplementary files or are available from the corresponding author upon request. The

source data underlying Figs. 1g and 3b, and Supplementary Figs. 9a–d are provided as a Source Data file.

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

## Acknowledgements

We thank A. Murphy, A. Middleton, and D. Wells for helpful and stimulating discussions on our data. We also thank S. Kitto for help with examining shoot-root interactions. We would also like to thank the following for seeds: J. Reed for *shy2-2*; B. Bartel for *iaa28-1*. K.H. and M.J.B. acknowledge the support of the Biotechnology and Biological Sciences Research Council (BBSRC) and Engineering and Physical Sciences Research Council (EPSRC) funding to the Center for Plant Integrative Biology (CPIB) BB/D019613/1; and K.H. from the BBSRC Professorial Research Fellowship funding to M.J.B. grant BB/G023972/1; and Royal Society-Wolfson Merit Award to M.J.B. This research was partially supported by the grants provided by the University of Delaware (UD) Graduate Fellowship to R.S. and by the grants from the National Science Foundation (IOS 1457121 and MCB 1820103) and from the IDeA program of the National Institute of General Medical Sciences (P30 GM103519) and UD College of Agriculture and Natural Resources Seed Grant to J.-Y.L.

## Author contributions

R.S. produced major experimental data, and R.S. and J.-Y.L. designed the experiments and wrote the manuscript. X.W. helped collecting LAX3 images in a time course and K.H. carried out ChIP assays. B.-C.Y. designed expression vector system and J.C. helped with image processing. During revision, J.-Y.L. performed CF loading with A.N.'s help collecting confocal images, and T.T. helped with supplementary RT-PCR. M.J.B. helped with data interpretation and manuscript editing.

## Competing interests

The authors declare no competing interests.
