## [Peer Review File · Nature Communications]

Reviewers' comments:

Reviewer #1 (Remarks to the Author):

In the submitted manuscript titled "Auxin-dependant spatiotemporal control of plasmodesmata regulator PDLP5 creates a negative feedback loop modulating lateral root emergence" by Sager et al., the authors present evidence that PDLP5, a plasmodesmata-associated receptor-like protein, restricts cell-to-cell connectivity in the LRP-overlying cells and control LRP emergence. From their data, the authors further propose that PDLP5 spatiotemporal expression is controlled by auxin and that PDLP5 up-regulation would in turn restrict symplastic connectivity and help to focus auxin distribution. The Ms. is very well written in a concise and logical manner, the figures presentation is of high quality, the results include controls, and overall contribute to support the main conclusions drawn by the authors.

In my opinion the work from Sager et al. represents an important advance in the field of cell-to-cell communication and organ growth in plants. The Ms. underpins and further highlights previous studies from the Maule's lab on the important of plasmodesmata for lateral root primordia and root branching, by showing that modulating symplastic connectivity around LRP is important for regulating the primordia emergence. This is an interesting paper that will benefit the field, provided the authors address the following major concerns.

1- Sager et al. propose that auxin regulates PDLP5 expression through ARF19. They support their claim by showing that PDLP5 promoter activity is increased in *shy2-2* mutant and repressed in *iaa28-1* mutant. They further used ChIP assays to show that ARF19 directly binds PDLP5 promoter and regulate PDLP5 expression. However, with the dominant mutant *SHY2-2* suppressing ARFs in the *shy2-2* mutant endodermal cells, how could ARF19 bypass *SHY2* and induce PDLP5 expression?

2- The authors only present indirect evidence that PDLP5 restricts cell-to-cell connectivity in cells overlying LRP. They do not demonstrate that symplastic connectivity and/or callose deposition is indeed modified locally in the relevance cell layers and through out LRP formation. For instance, the authors show that induced PDLP5 expression increases callose accumulation but in the primary root tip (and not around LRP) and only slightly (Fig.2 C – see further comment about callose quantification). Same goes with the GFP cell-to-cell connectivity tests using *CASP1:GFP* and inducible PDLP5 lines which do not monitor symplastic movement in the relevant tissues. So from their present data, the authors cannot really conclude that "auxin-dependant PDLP5 expression stimulates the formation of a temporary symplastic domain in LRP-overlying cells".

Another reason for raising this point is that previous work by Benitez-Alonso et al. (*Dev cell* 26, 136-147, 2013) actually showed that GFP expressed under *SUC2* promoter could move freely in the stele, endodermis, cortex and epidermis at stage I-II of LR development. Yet, in the present work, the authors clearly show that PDLP5 is strongly expressed at these stages (I-II) in the endodermis overlying the nascent primordium. This somehow contradicts Sager et al. claim that PDLP5 expression restrict cell-to-cell communication in LRP overlying cells, at least not at this stage.

As I understood these are challenging experiments but the authors should try to monitor symplastic movement and callose in the relevant tissue using for instance *CASP1:GFP*; *SUC2:GFP* for early stages or CFDA and immunolocalisation for callose (for instance in PDLP5 over-expressing lines where callose should be more easily detected).

3-In Fig. 2C, from the presented confocal images, it actually looks as if only cell plates and not plasmodesmata were labelled by aniline blue staining. Immunolocalisation would be far more more accurate and sensitive for callose quantification.

4- I think it would strengthen the present paper if the authors could confirm PDLP5 localisation to plasmodesmata in LRP zone using for instance immunogold labelling or co-labelling with markers.

5- In Supplementary figure3, the authors use a different experimental set up to investigate hormone-dependant induction of PDLP5 expression whether it is GUS or RT-PCR analyses. For GUS, 100 nM and 1 μ M of NAA was used on 7 days seedlings and 9 hours treatment, whereas for RT-PCR 5 μ M of NAA was used on 9 days seedlings and 4 hours treatment. For consistency, the two sets of experiments should be conducted in the same conditions. Also for RT-PCR it would be nice to see additional reference/house keeping genes and not only UBQ. Last, are the authors sure that only PDLP5 expression and not other PDLP members is detected in the RT-PCR

6- Supplementary Table 3 and 4 are missing from the Suppl data

Reviewer #2 (Remarks to the Author):

This paper introduces the idea that symplastic isolation may play an important role in lateral root development. A membrane bound protein PDLP5 that has previously been shown to localize to plasmodesmata is investigated in terms of its effect on root branching. A movement assay confirms that expression of PDLP5 in the root endodermis is capable of limiting movement of soluble GFP. Loss of *pdlp5* activity increases the number of tertiary lateral roots formed, without having much if any effect on secondary root production. Loss of function also leads to a modest increase in lateral root emergence. Overexpression leads to a set of inverse phenotypes. Thus, PDLP5 represses root branching both in terms of number and rate of emergence. The consideration of plasmodesmatal transport as a regulator of LR development is new and welcome; the authors deserve kudos for the idea. Some issue with the execution remain and are detailed below.

Major comments

The PDLP5 expression in WT shown in Figure 1e (and also somewhat in Supp Fig 1) is concentrated in the LRP; this looks more like DR5:GUS than it does the pattern described for PDLP5 in Figure 1a. A more comprehensive description of the WT pattern that includes all of the patterns seen needs to be given at the start of the paper.

In Figure 1 g, it appears that NAA causes expression of PDLP5 to move from the outside (epidermal cells?) toward the interior of the root. An epidermal starting location does not match what is shown in Fig 1a and Fig 2a, and that difference should be addressed. Also, a shift in location is more complex than a simple increase in abundance and should be clearly described in the text.

The concluding model states that PDLP5 expression closes plasmodesmata in cells overlying a lateral root primordium, resulting in a decrease in the amount of auxin in the cell, which leads to a reduction in LAX3 expression. PDLP5 expression is shown to affect the number of cortical cells in which the DR5 promoter leads to expression, and the intensity of DR5-driven expression in the tips of the LRP themselves (Supp Fig 10), but the argument is that the level of auxin is reduced in overlying cells. Given the centrality of the expected reduction in DR5 intensity to the final model, and the existence of the necessary lines, the effect of PDLP5 on the intensity of DR5 expression in overlying cells should be directly investigated. (The PDLP5-induced decrease in LAX3 expression could be indirect.)

The statement that JA and BAP do not affect PDLP5 expression does not match the images shown in the supplement. JA appears to increase expression in outer cell files in what (approximately, as far as can be seen in this one photo) aligns with the elongation zone. BAP appears to limit the vascular expression to a region close to the root tip.

The first paragraph of discussion asserts that PDLP5 expression ensures an optimal rate of LR emergence. What is an optimal rate? How would one know? The data show that PDLP5 decreases LR initiation and slows LR emergence. These findings should be clearly summarized. After that, any additional speculation clearly labeled as such. Similarly, the second paragraph of the discussion should start with a statement making it clear that the model is what is being described.

Moderate comments

I don't understand this section: "A developing LRP is thought to sense mechanical resistance in the overlying En cells and abort if turgor of these cells does not decrease as it grows [Vermeer et al. 2014]. Therefore, it is an interesting possibility that PD closure in LRP-overlying cells may be necessary for building up the turgor pressure needed for normal progression of LRP emergence." The wording would make sense if the turgor that needs to increase is the turgor of the LRP (agreeing with the expression domain shown in figure 1 e), but it seems that the turgor in the overlying cells needs to decrease, not increase.

Is the width of the LRP slightly more elongated in *pdlp5-1* than in WT, and slightly narrower in *PDPLPOE*? (See Supp Fig 11, left panels). If so, this could have an impact on the number of cortical cells that stain, or vice versa.

When shoots were removed and PDLP5 expression fell, was LR emergence faster? Possibly not as missing the entire shoot could have multiple effects, but it seems important to mention the answer as the lower level of PDLP5 would argue for a higher rate of emergence.

The difference in effect on secondary and tertiary roots is fascinating. Although the reasons behind such a difference are not yet clear, it would be wonderful if the discussion could highlight this distinction.

Editorial comments

In Figure 1d, what is the basis for the location of the arrow heads? If higher mag photos showed these locations to contain LRP, it seems good to include those in the supplement. If there is no direct evidence, the arrowheads should be removed.

Please indicate which ecotype was used as the WT control and for the mutants in the legend of figure 1e. Also, what is the evidence that the *shy2-2* LRP were aborted?

It is not clear what is shown in Figure 1g. Figure 2 includes clear illustrations; something similar along with more description in the legend should be included in Figure 1.

In Figure 2A, the large number of arrows and other marks makes it difficult to see the actual signal. Please remove the arrowheads from the photographs. The carat and dotted line can probably stay. Use the legend to point out that the signal appears as small blue or white dots, depending on intensity.

In Figure 2B, I miss a scale bar.

In Figure 4b, replacing the title with one that draws attention to the fact that it is DR5 expressing cells (not the total number of cells per se) would increase clarity.

The paper would reach a wider audience if some changes in writing were implemented:

Especially at the start, but also in other places, try to direct the reader to one of the main figures, and then follow up with references to the supplement. Pointing to the supplement first may cause frustration.

Spell out PD (page 4 subhead and throughout the text). People aren't used to seeing the term in this context, and it's easier to read and understand a word rather an abbreviation.

Define pER8 at first use in results section

State (and reference) that aniline blue stains callose

Reviewer #3 (Remarks to the Author):

Lateral root primordia need to pass through several overlying layers of cells to develop into lateral roots, and the process of lateral root emergence has been shown to be regulated independent of the process of lateral root initiation. Nevertheless, much about the process of emergence and its regulation remains mysterious. Here, the authors extend a previous finding by Benitez-Alfonso et al (2013) who demonstrated that symplastic connectivity between cells in the LRP is regulated during their development, and that regulation of callose levels determines this connectivity through its action of plasmodesmatal permeability. Interestingly, that study demonstrated the role of 2 b-glucanases, expressed in early LRP, in reducing callose levels and negatively regulating lateral root initiation (but not emergence). In the present study, the authors argue that a different protein, a receptor-like transmembrane protein called PDLP5 that was previously shown to increase callose formation, has a negative effect on LRP emergence. The opposite direction of the effects of callose modulation likely reflects the different expression patterns of the callose modulating genes examined, and the different roles of the cells expressing these genes in the lateral root formation process. Why callose, leading to symplastic isolation and reduced cell-cell communication via PDs should affect later stages of lateral root formation is addressed in the current paper.

The authors demonstrate convincingly that the plasmodemata-localized gene PDLP5 is expressed in the cells overlying the developing LRP, turning on progressively in endodermis, cortex and epidermis as the LRP emerge. Expression requires shoot-derived auxin. In a beautiful experiment, the authors demonstrate that estradiol-induced PDLP5 expression leads to increased callose accumulation and limits the movement of a mobile GFP reporter from pericycle to endodermis and other tissues, showing that PDLP5 can restrict cell-cell communication from the primordia to the overlying tissues. PDLP has a negative effect on lateral root emergence, suggesting that limiting cell-cell communication between primordia and overlying cells may constrain emergence. The authors ask whether this could act via auxin, consistent with a previous model from this lab that auxin movement between overlying cells promotes emergence. In support of this idea, they claim that the PDLP over expression leads to a decrease in the number of overlying cortex cells expressing an auxin reporter, but this result is the only one that is a bit unconvincing, as opposed to everything else shown in this well constructed study. Despite this weakness, there is clearly a connection to auxin as expression of the auxin-induced gene LAX3 is inversely correlated with PDLP5 expression, and the increased expression in the pdlp5 mutants is in a wider domain of cells.

Together, these results add significantly to the data that can be brought to bear on the mystery of lateral root emergence. They suggest that LRP emergence may be constrained by PDLP5-regulated

communication between LRP and overlying cells. A negative feedback loop emerges in which shoot-derived auxin induces PDL5 in cells overlying the LRP; this induces PD closure which restricts the movement of further auxin out of the primordia and into overlying cells. Downstream genes such as LAX3 are therefore not induced in these overlying cells, further restricting auxin distribution in overlying tissues. How this regulatory circuit ultimately provides correct regulation of LRP emergence is still a bit foggy, and the authors ideas about LRP compressing the overlying cells and the need for plasmodesmata to function in resulting turgor are interesting but quite speculative.

This a well written paper with rigorous experiments that are consistent with the interpretations provided by the text. All the figures and data are very convincing with the sole exception cited above. The data and ideas are novel and move the field forward, although not far enough forward to have a transformative effect.

Authors' responses to reviewers' comments:

Reviewer #1 (Remarks to the Author):

In the submitted manuscript titled "Auxin-dependant spatiotemporal control of plasmodesmata regulator PDLP5 creates a negative feedback loop modulating lateral root emergence" by Sager et al., the authors present evidence that PDLP5, a plasmodesmata-associated receptor-like protein, restricts cell-to-cell connectivity in the LRP-overlying cells and control LRP emergence. From their data, the authors further propose that PDLP5 spatiotemporal expression is controlled by auxin and that PDLP5 up-regulation would in turn restrict symplastic connectivity and help to focus auxin distribution. The Ms. is very well written in a concise and logical manner, the figures presentation is of high quality, the results include controls, and overall contribute to support the main conclusions drawn by the authors.

In my opinion the work from Sager et al. represents an important advance in the field of cell-to-cell communication and organ growth in plants. The Ms. underpins and further highlights previous studies from the Maule's lab on the important of plasmodesmata for lateral root primordia and root branching, by showing that modulating symplastic connectivity around LRP is important for regulating the primordia emergence. This is an interesting paper that will benefit the field, provided the authors address the following major concerns.

1. Sager et al. propose that auxin regulates PDLP5 expression through ARF19. They support their claim by showing that PDLP5 promoter activity is increased in *shy2-2* mutant and repressed in *iaa28-1* mutant. They further used ChIP assays to show that ARF19 directly binds PDLP5 promoter and regulate PDLP5 expression. However, with the dominant mutant SHY2-2 suppressing ARFs in the *shy2-2* mutant endodermal cells, how could ARF19 bypass SHY2 and induce PDLP5 expression?

Response: Orosa-Puente et al (2018) recently reported in *Science* that the SHY2 interaction with ARF activating proteins is unusual compared to other Aux/IAA repressor proteins. This paper demonstrated that ARF7 requires to be modified with SUMO in order to interact with SHY2/IAA3 (via a SUMO-interaction motif; SIM). Given these results, we propose ARF19 bypasses SHY2-2 repression by not being SUMOylated. To better explain this point, we have added the following sentence to the results section.

*"The differential regulatory effect of *iaa28-1* and *shy2-2* may reflect the recent report (Orosa et al, 2018) that SHY2/IAA3 requires ARF targets to be SUMOylated before it can interact and repress their transcriptional activity."*

2a. The authors only present indirect evidence that PDLP5 restricts cell-to-cell connectivity in cells overlying LRP. They do not demonstrate that symplastic connectivity and/or callose deposition is indeed modified locally in the relevance cell layers and throughout LRP formation. For instance, the authors show that induced PDLP5 expression increases callose accumulation but in the primary root tip (and not around LRP) and only slightly (Fig.2 C – see further comment about callose quantification). Same goes with the GFP cell-to-cell connectivity tests using CASP1:GFP and inducible PDLP5 lines which do not monitor symplastic movement in the relevant tissues. So from their present data, the authors cannot really conclude that "auxin-dependant PDLP5 expression stimulates the formation of a temporary symplastic domain in LRP-overlying cells".

Response: We toned down our conclusion by softening the statements made in the introduction and results sections as follows.

“In the current study, we report that our data suggest that auxin-dependent PDLP5 expression stimulates the formation of a temporary symplasmic domain in LRP-overlying cells”.

“This result suggests that PDLP5 is capable of restricting PD permeability in root cells.”

Another reason for raising this point is that previous work by Benitez-Alonso et al. (Dev cell 26, 136-147, 2013) actually showed that GFP expressed under SUC2 promoter could move freely in the stele, endodermis, cortex and epidermis at stage I-II of LR development. Yet, in the present work, the authors clearly show that PDLP5 is strongly expressed at these stages (I-II) in the endodermis overlying the nascent primordium. This somehow contradicts Sager et al. claim that PDLP5 expression restrict cell-to-cell communication in LRP overlying cells, at least not at this stage.

Response: We appreciate the scrutiny of this comment comparing study by Benitez-Alonso et al. The data corresponding to the reviewer’s point is included as Figure 1 in Benitez-Alonso et al (2013), which the authors described that “In the basal meristem and early LR forming region of 6-day-old roots, we detected GFP signal in the stele, endodermis, cortex, and epidermis suggesting that symplastic connectivity between the phloem and the outer cell layers is maintained in this region (Figure 1B)”.

It is plausible that the GFP detected in the endodermis of the SUC2:GFP roots moved into that tissue prior to the symplasmic isolation that we hypothesize occurs during early LRP development in stages 1-3 (0-12 hrs). GFP can remain quite stable within tissue for many hours after expression. Therefore, our data does not necessarily conflict with Benitez-Alfonso et al; if the GFP (expressed by SUC2:GFP) was within the overlying endodermal tissue prior to PDLP5 expression during LRP emergence, the signal would remain even as symplasmic isolation occurred in those cells. A better test would be to add an ectopic fluorescent protein/reporter after the LRP developmental process has begun. In fact, an experiment similar to this was performed by Oparka et al (1995, J ExB 46:187-197). They showed, using phloem CF dye loading, that the symplastic dye moved into unemerged LRP at early stages, but the authors did not make a note of the dye penetrating any cells located outside a developing LPR.

2b. As I understood these are challenging experiments but the authors should try to monitor symplastic movement and callose in the relevant tissue using for instance CASP1:GFP; SUC2:GFP for early stages or CFDA and immunolocalisation for callose (for instance in PDLP5 over-expressing lines where callose should be more easily detected).

Response: As appreciated by the reviewer, quantifying PD callose levels and PD permeability within the internal 2-3 root cells is technically very challenging. Therefore, we devised the alternative approaches we explain in the paper, such as utilizing CASP1:GFP in conjunction with inducible PDLP5 to validate the functionality of PDLP5 as a PD regulator in roots, as already demonstrated in shoots*.

We revised the following text to make this point clearer:

“Since directly assessing PD permeability across a few internal root cells is not currently possible, we designed a new experimental set-up that would allow us to evaluate PDL5 function indirectly in restricting cell-to-cell movement across the En cell layer.”

*We have previously shown using co-localization, correlative light and electron microscopy, and immunogold labeling that PDL5-GFP is exclusively localized at both primary and secondary PD (see Figures 2,3,4, and 5 by Lee et al., Plant Cell (2011, 23:3353-3373). In the same paper and follow-up papers (Wang et al, 2013 in Plant Cell; Cui and Lee, 2016 in Nature Plants; Lim et al, 2016 in Cell Host & Microbe), we have also shown that PDL5 induction reduces PD permeability via a callose-dependent mechanism.

3. In Fig. 2C, from the presented confocal images, it actually looks as if only cell plates and not plasmodesmata were labelled by aniline blue staining. Immunolocalisation would be far more accurate and sensitive for callose quantification.

Response: When roots are imaged by confocal for aniline blue staining, the cross walls are most distinct because the punctate spots are collected in horizontal plane. In contrast, the punctate spots in radial and tangential walls, which are curved walls, would appear singularly*. As a result, those signals are few in number and not easily detectable, especially under the imaging conditions used in our experiment as detailed below.

The aniline blue imaging was conducted using multiphoton excitation microscopy. Individual plasmodesmata were not resolved because these images only have a 378 nm lateral (XY) resolution and 1076 nm axial (Z) resolution. A higher resolution 3D Z-stack starting from the epidermal layer may have better revealed individual callose-marked punctae in the outer tissues and along the cross walls. However, higher resolution imaging would have resulted in more photobleaching and would have prevented accurate quantification. Please note also that fluorescence of aniline blue is easily photo-bleachable. Our image acquisition settings were optimized for minimal photobleaching (low laser power and magnification) for quantification purposes.

We are not objecting to the reviewer’s point that immunolocalization could be more accurate and sensitive for callose quantification. However, immunolocalization has its own set of challenges that includes antibody penetration and a related cell wall digestion step. In our hands, that step increases variability making it more difficult to quantify. Furthermore, repeating these experiments with immunolocalization would not change our conclusion, and in our experience, among all the trials we attempted, the most reproducible quantification of overall PD callose levels in Arabidopsis was achievable using aniline-blue staining of live cells.

*For additional examples, please refer to Ross-Elliott et al., eLife (2017) for the aniline-blue stained root images showing signals at cross walls. Also, for the PD at the cross wall vs. other walls, please see our revised main Figure 1e (moved to 1e from 1g in the revision) as provided in response to Reviewer #2’s Editorial comments #3.

4. I think it would strengthen the present paper if the authors could confirm PDL5 localization to plasmodesmata in LRP zone using for instance immunogold labelling or co-labelling with markers.

Response: PDL5-GFP is a very strong, reliable marker of PD localization. We have already published PD localization of PDL5—please see earlier response 2.

5. In Supplementary figure 3, the authors use a different experimental set up to investigate hormone-dependent induction of PDLP5 expression whether it is GUS or RT-PCR analyses. For GUS, 100 nM and 1 μ M of NAA was used on 7 days seedlings and 9 hours treatment, whereas for RT-PCR 5 μ M of NAA was used on 9 days seedlings and 4 hours treatment. For consistency, the two sets of experiments should be conducted in the same conditions. Also for RT-PCR it would be nice to see additional reference/housekeeping genes and not only UBQ. Last, are the authors sure that only PDLP5 expression and not other PDLP members is detected in the RT-PCR.

Response: The different seedling ages and treatment times were used to maximize the effectiveness of the experiments in each case. In our experimental conditions, we found that 7-day-old seedlings are the best for imaging GUS in the roots. For RNA extraction, we chose 9-day-old roots because they are larger and thus easier to flash freeze in liquid nitrogen and collect. The incubation time of 4-hours was chosen because we needed an early time point where PDLP5 expression would be induced by the hormone, but before any potential stress-related side effects from the treatments could occur, because PDLP5 can be stress-induced in certain circumstances. Furthermore, we tested later time points as well, and PDLP5 RNA responsiveness was detected at multiple treatment times and different seedling ages as well.

We used primers that were confirmed gene-specific in our previous publication (Lee et al., 2011).

6. Supplementary Table 3 and 4 are missing from the Suppl data.

Response: We apologize for mistakenly omitting those tables. They are included in the resubmission.

Reviewer #2 (Remarks to the Author):

This paper introduces the idea that symplastic isolation may play an important role in lateral root development. A membrane bound protein PDLP5 that has previously been shown to localize to plasmodesmata is investigated in terms of its effect on root branching. A movement assay confirms that expression of PDLP5 in the root endodermis is capable of limiting movement of soluble GFP. Loss of pdlp5 activity increases the number of tertiary lateral roots formed, without having much if any effect on secondary root production. Loss of function also leads to a modest increase in lateral root emergence. Overexpression leads to a set of inverse phenotypes. Thus, PDLP5 represses root branching both in terms of number and rate of emergence. The consideration of plasmodesmatal transport as a regulator of LR development is new and welcome; the authors deserve kudos for the idea. Some issue with the execution remain and are detailed below.

Major comments:

1. The PDLP5 expression in WT shown in Figure 1e (and also somewhat in Supp Fig 1) is concentrated in the LRP; this looks more like DR5:GUS than it does the pattern described for PDLP5 in Figure 1a. A more comprehensive description of the WT pattern that includes all of the patterns seen needs to be given at the start of the paper.

Response: We would like to clarify that we never detected PDLP5 expression inside the LRP, as we have shown in Figure 1a, top left panel in this paper. Also note that PDLP5 does not express in the primary root meristem. In Figure 1E, the angle of the image is such that the non-GUS-stained LRP is located below the overlying GUS-stained cells, so the blue coloration overlaps. GUS stain sometimes seeps slightly into neighboring cells where the promoter is inactive (this is apparent in the magnified PDLP5-GUS image in Supp Fig 1). GUS is a useful tool, but because it does sometimes leak, we confirmed multiple times conclusively using PDLP5:ER-YFP and PDLP5pro:PDLP5-GFP that expression within the LRP never occurs.

2. In Figure 1g, it appears that NAA causes expression of PDLP5 to move from the outside (epidermal cells?) toward the interior of the root. An epidermal starting location does not match what is shown in Fig 1a and Fig 2a, and that difference should be addressed. Also, a shift in location is more complex than a simple increase in abundance and should be clearly described in the text.

Response: The PDLP5 endogenous expression pattern (Figs 1A, 2A) is from endogenous auxin moving from the shoot to the primary root, and then from phloem to outer tissues via the new LRP (as described by Swarup et al, 2008, Nature Cell Biology). In contrast, exogenous membrane-permeable NAA treatment is first detected in the outer tissues, as the chemical enters root tissues. Exogenous NAA treatment induces PDLP5 expression in tissues outside those it is normally induced by endogenous auxin. Our NAA experiment was simply to show that ectopic auxin can induce PDLP5 protein.

3. The concluding model states that PDLP5 expression closes plasmodesmata in cells overlying a lateral root primordium, resulting in a decrease in the amount of auxin in the cell, which leads to a reduction in LAX3 expression. PDLP5 expression is shown to affect the number of cortical cells in which the DR5 promoter leads to expression, and the intensity of DR5-driven expression in the tips of the LRP themselves (Supp Fig 10), but the argument is that the level of auxin is reduced in overlying cells. Given the centrality of the expected reduction in DR5 intensity to the final model, and the existence of the necessary lines, the effect of PDLP5 on the intensity of DR5 expression in overlying cells should be directly investigated. (The PDLP5-induced decrease in LAX3 expression could be indirect.)

Response: We attempted to measure the intensity of the DR5:3VENUS nuclei at the same time as counting the number in the overlying cortical cells. However, the variance in the fluorescence intensity of DR5:3VENUS was high in WT plants, and was greater than any differences between WT, *pdlp5-1*, and PDLP5OE. We believe that this variance was due to variation in imaging depth that resulted in variation in fluorescence intensity drop-off from imaging deeper in the tissue. Related issues included differences in the root and LRP sizes and variation in mounting. DR5:GUS strongly stains LRP and root tips, but does *not* consistently stain LRP-overlying cells.

For the purposes of this paper, our model focuses mostly on how auxin movement into LRP-overlying cells during the separation process is delayed by PDLP5, which in turn delays the timing of emergence. We do not believe that auxin concentration is reduced drastically by PDLP5, though a more sensitive auxin-detecting technique could perhaps be used in a future study.

4. The statement that JA and BAP do not affect PDLP5 expression does not match the images shown in the supplement. JA appears to increase expression in outer cell files in what (approximately, as far as

can be seen in this one photo) aligns with the elongation zone. BAP appears to limit the vascular expression to a region close to the root tip.

Response: JA and BAP do cause slight changes in PDLP5pro:GUS expression, as JA and ABA do to DR5:GUS expression. However, we were focusing on describing a stark change (strong induction) similar to the SA response. Furthermore, for BAP, there could possibly be a change in the zone of PDLP5 expression in the protoxylem, due to auxin/cytokinin antagonism. We revised the pertinent text to clarify this point as follows:

“Other hormones such as cytokinin, jasmonic acid, and abscisic acid, had little to no effect on PDLP5 induction in roots, although cytokinin seems to reduce the area of PDLP5pro:GUS expression within the protoxylem.”

5. The first paragraph of discussion asserts that PDLP5 expression ensures an optimal rate of LR emergence. What is an optimal rate? How would one know? The data show that PDLP5 decreases LR initiation and slows LR emergence. These findings should be clearly summarized. After that, any additional speculation clearly labeled as such. Similarly, the second paragraph of the discussion should start with a statement making it clear that the model is what is being described.

Response: The reviewer is correct that “Optimal” is not the best word choice. We intended it to convey the “normal” rate, the average time it would take an LRP to develop and emergence in the same conditions with no outside variables. These are good writing suggestions. We revised this Discussion section in our manuscript to read as follows:

“Our study reveals a novel negative feedback loop linking the plasmodesmal regulator PDLP5 with auxin in LRP-overlying cells. This regulatory circuit functions to symplasmically isolate LRP-overlying cells while ensuring organ emergence occurs at a normal rate. Starting soon after founder cell division begins, auxin induces spatiotemporal expression of PDLP5 in the LRP-overlying cells, where PDLP5 localizes to plasmodesmata. PDLP5 induction in roots stimulates callose deposition and closes plasmodesmata, inhibiting cell-to-cell movement of En-expressed GFP into outer Co cells. PDLP5 negatively feedback-regulates auxin in LRP-overlying cells, altering the timing and number of overlying cells expressing auxin marker DR5 and LAX3. Lastly, lack of PDLP5 increases higher-order LRP development and emergence rate.”

Also, the second paragraph of the Discussion section is revised to start with the following sentence.

“We summarize the PDLP5-auxin functional relationship in our model (Fig. 4h) and integrate key factors, such as the ARF7/19 module and cell turgor, which are vital for normal LRP development and emergence.”

Moderate comments:

I don't understand this section: “A developing LRP is thought to sense mechanical resistance in the overlying En cells and abort if turgor of these cells does not decrease as it grows Vermeer et al. 2014. Therefore, it is an interesting possibility that PD closure in LRP-overlying cells may be necessary for building up the turgor pressure needed for normal progression of LRP emergence.”

The wording would make sense if the turgor that needs to increase is the turgor of the LRP (agreeing with the expression domain shown in Figure 1e), but it seems that the turgor in the overlying cells needs to decrease, not increase.

Response: We apologize for any confusion and have amended the sentence to read as follows:

“A developing LRP is thought to sense mechanical resistance in the overlying En cells, and abort if turgor of these cells does not decrease as the growing LRP pushes against them²⁵. Therefore, it is an interesting possibility that PD closure in LRP-overlying cells may be necessary for modifying the turgor pressure needed for normal progression of LRP emergence.”

Is the width of the LRP slightly more elongated in *pdlp5-1* than in WT, and slightly narrower in *PDLPOE*? (See Supp Fig 11, left panels). If so, this could have an impact on the number of cortical cells that stain, or vice versa.

Response: The WT and *pdlp5-1* LRP sizes have always been similar in our observations. *PDLPOE* LRP have delayed emergence, and can thus appear smaller when imaged at the same time points as WT and *pdlp5-1*. However, the time point (27 hours post-initiation) shown in this image was chosen because the relative location of the LRP (moving into the cortex) would be comparable between WT, *pdlp5-1*, and *PDLPOE*, thus allowing comparisons of DR5-3VENUS nuclear counts in the overlying cortical cells.

When shoots were removed and *PDLP5* expression fell, was LR emergence faster? Possibly not as missing the entire shoot could have multiple effects, but it seems important to mention the answer as the lower level of *PDLP5* would argue for a higher rate of emergence.

Response: We appreciate this point. While we did not specifically test LRP emergence rate in shoot-removed seedlings, it is unlikely to have increased LRP emergence; as reported by Bhalerao et al (2002, Plant J. 29:325-332), removal of the aerial tissue at an early seedling age caused a drastic reduction in LRP development. Furthermore, we agree with the reviewer’s comment that the loss of shoot-to-root auxin would impact more factors than *PDLP5* alone.

The difference in effect on secondary and tertiary roots is fascinating. Although the reasons behind such a difference are not yet clear, it would be wonderful if the discussion could highlight this distinction.

Response: We agree that this is a highly intriguing phenotype. We added the following paragraph in the Discussion section.

*“Another interesting question our data raise is how auxin-dependent induction of *PDLP5* during LR emergence might regulate root architecture, altering LR branching patterns. Both primary and secondary root lengths as well as lateral root numbers are reduced in *PDLP5-OE* seedlings, while secondary root length and tertiary root numbers are increased without affecting the primary root length in *pdlp5-1* mutants. It is known that plant root architecture changes in response to internal and external nutrient states, and certain nutrients are linked to specific root morphological and architectural modifications²⁷. For example, while severe nitrogen deficiency inhibits overall root growth, mild nitrogen deficiency stimulates lateral root emergence and elongation²⁸. Notably, this emergence and elongation is driven by auxin accumulating in later stage LRP, at a similar time to when the first differences between WT*

and pdlp5-1 root emergence rate can also be observed. Future investigation may uncover a yet unknown signaling mechanism that suppresses PDLP5 in the roots, so that when such a nutrient foraging program is activated, greater lateral root emergence and elongation can occur.”

Editorial comments

1. In Figure 1d, what is the basis for the location of the arrow heads? If higher mag photos showed these locations to contain LRP, it seems good to include those in the supplement. If there is no direct evidence, the arrowheads should be removed.

Response: This is a very good suggestion. We have added a close-up image in Supplemental Figure 3b.

2. Please indicate which ecotype was used as the WT control and for the mutants in the legend of figure 1e. Also, what is the evidence that the shy2-2 LRP were aborted?

Response: The ecotype Col-0 was used as WT, shy2-2 was Ler, iaa28-1 was Ws. This information is added to the legend. We examined shy2-2 roots under the microscope and could find few to no LRP past stage 5. Prior studies into shy2 mutant phenotypes [Goh et al 2012] have supported our observations as well.

We changed “aborted” to “unemerged” and added a sentence in the text for clarification as follows.

“In contrast, PDLP5 expression was strongly upregulated in En cells above the large number of unemerged LRP in shy2-2 roots (Fig. 1f; Supplementary Fig. 4). We examined shy2-2 roots under the microscope and could find few to no LRP past stage V, as reported elsewhere²⁰.”

3. It is not clear what is shown in Figure 1g. Figure 2 includes clear illustrations; something similar along with more description in the legend should be included in Figure 1.

Response: We have included a cartoon and revised the figure panel name and legend as follows.

“ e) Induction of PDLP5-GFP in 10 μ M NAA-treated PDLP5pro:PDLP5-GFP;pdlp5-1 roots. Mock-treated epidermal cells exhibit non-specific background green fluorescence within and at the outer surface of the cells exposed to the media. In contrast, NAA treatment induces PDLP5-GFP expression, which accumulate as punctate fluorescent signals at plasmodesmata in cross walls between Co cells (arrows) and tangential walls between Co and Epi junctions (darts). Cartoons show absence and presence of PDLP5-GFP signals at plasmodesmata (red dots) in root cross-sectional and longitudinal representations of confocal images.”

4. In Figure 2A, the large number of arrows and other marks makes it difficult to see the actual signal. Please remove the arrowheads from the photographs. The carat and dotted line can probably stay. Use the legend to point out that the signal appears as small blue or white dots, depending on intensity.

Response: We are thankful for this comment that these edits improve the clarity of the images. We removed arrowheads pointing to punctate PDLP5-GFP signals as suggested by the reviewer and changed double-direction arrows to empty arrowheads, pointing to separated cell walls. We also added labels to the cartoon.

We have also revised the figure legend as follows.

“Plasmodesmata marked by punctate PDLP5-GFP signal (in cartoons, red dots) can be seen at the cell junctions. PDLP5-GFP signals vary in color ranging from blue to green, yellow, and white depending on fluorescence intensity as indicated in calibration bar of the fluorescence intensity included on the right. The dashed arcs are positions of LRP; open arrowheads, separated cell walls; carets, strong PDLP5-GFP signal in cell walls just prior to separation.”

5. In Figure 2B, I miss a scale bar.

Response: A scale bar added.

6. In Figure 4b, replacing the title with one that draws attention to the fact that it is DR5 expressing cells (not the total number of cells per se) would increase clarity.

Response: Title changed to “Overlying Co cells expressing DR5:3VENUS”.

7. The paper would reach a wider audience if some changes in writing were implemented. Especially at the start, but also in other places, try to direct the reader to one of the main figures, and then follow up with references to the supplement. Pointing to the supplement first may cause frustration.

Response: We have revised the order of figure appearance, pointing to the main figure first wherever possible throughout the text.

1. Spell out PD (page 4 subhead and throughout the text). People aren’t used to seeing the term in this context, and it’s easier to read and understand a word rather an abbreviation.

Response: We spelled out PD in the revision throughout the text.

2. Define pER8 at first use in results section.

Response: We revised the text as follows.

“Therefore, we utilized the Arabidopsis transgenic line pER8:PDLP5 that we had described elsewhere¹⁴, which expresses PDLP5 under the control of an estradiol-inducible promoter, to assess if we could correlate ectopic PDLP5 induction and plasmodesmal callose levels in root tip cells where callose staining was possible.”

3. State (and reference) that aniline blue stains callose.

Response: The following sentence with a reference is added to the text.

“Aniline blue binds to callose to give a yellow fluorescence in ultraviolet light ²².”

We thank for these detailed editorial comments provided by reviewer #2. These comments were very helpful to make specific descriptions in the text clearer and more informative.

Reviewer #3 (Remarks to the Author):

Lateral root primordia need to pass through several overlying layers of cells to develop into lateral roots, and the process of lateral root emergence has been shown to be regulated independent of the process of lateral root initiation. Nevertheless, much about the process of emergence and its regulation remains mysterious. Here, the authors extend a previous finding by Benitez-Alfonso et al (2013) who demonstrated that symplastic connectivity between cells in the LRP is regulated during their development, and that regulation of callose levels determines this connectivity through its action of plasmodesmatal permeability. Interestingly, that study demonstrated the role of 2 b-glucanases, expressed in early LRP, in reducing callose levels and negatively regulating lateral root initiation (but not emergence). In the present study, the authors argue that a different protein, a receptor-like transmembrane protein called PDLP5 that was previously shown to increase callose formation, has a negative effect on LRP emergence. The opposite direction of the effects of callose modulation likely reflects the different expression patterns of the callose modulating genes examined, and the different roles of the cells expressing these genes in the lateral root formation process. Why callose, leading to symplastic isolation and reduced cell-cell communication via PDs should affect later stages of lateral root formation is addressed in the current paper.

The authors demonstrate convincingly that the plasmodesmata-localized gene PDLP5 is expressed in the cells overlying the developing LRP, turning on progressively in endodermis, cortex and epidermis as the LRP emerge. Expression requires shoot-derived auxin. In a beautiful experiment, the authors demonstrate that estradiol-induced PDLP5 expression leads to increased callose accumulation and limits the movement of a mobile GFP reporter from pericycle to endodermis and other tissues, showing that PDLP5 can restrict cell-cell communication from the primordia to the overlying tissues. PDLP has a negative effect on lateral root emergence, suggesting that limiting cell-cell communication between primordia and overlying cells may constrain emergence. The authors ask whether this could act via auxin, consistent with a previous model from this lab that auxin movement between overlying cells promotes emergence. In support of this idea, they claim that the PDLP over expression leads to a decrease in the number of overlying cortex cells expressing an auxin reporter, but this result is the only one that is a bit unconvincing, as opposed to everything else shown in this well constructed study. Despite this weakness, there is clearly a connection to auxin as expression of the auxin-induced gene LAX3 is inversely correlated with PDLP5 expression, and the increased expression in the pdlp5 mutants is in a wider domain of cells.

Response: We appreciate the supportive and constructive comments made by reviewer 3, particularly that we have provided a series of synergistic (e.g. imaging and genetic) experimental results which collectively support our model.

Together, these results add significantly to the data that can be brought to bear on the mystery of lateral root emergence. They suggest that LRP emergence may be constrained by PDLP5-regulated communication between LRP and overlying cells. A negative feedback loop emerges in which shoot-derived auxin induces PDLP5 in cells overlying the LRP; this induces PD closure which restricts the

movement of further auxin out of the primordia and into overlying cells. Downstream genes such as LAX3 are therefore not induced in these overlying cells, further restricting auxin distribution in overlying tissues. How this regulatory circuit ultimately provides correct regulation of LRP emergence is still a bit foggy, and the authors' ideas about LRP compressing the overlying cells and the need for plasmodesmata to function in resulting turgor are interesting but quite speculative.

Response: We have focused in the results section on making concrete conclusions based on our experimental data, and reserving the wider and more speculative mechanistic questions and implications of our findings (e.g. PD regulation on turgor regulation) in the Discussion section.

This a well written paper with rigorous experiments that are consistent with the interpretations provided by the text. All the figures and data are very convincing with the sole exception cited above. The data and ideas are novel and move the field forward, although not far enough forward to have a transformative effect.

Response: We appreciate the supportive and constructive comments made by reviewer 3 about the novelty of our reported findings and implications of our work.

We thank all Reviewers for their enthusiasm and constructive feedback. Inspired by many of your comments, we updated our model in the revision to help clarify our hypotheses about PDLP5 function.

Reviewers' comments:

Reviewer #1 (Remarks to the Author):

In their response, the authors have only partially replied to my concerns. In particular, given the centrality of the reduction of symplastic movement in their model, I still think the authors should provide experimental data demonstrating, instead of postulating, that at early stages of LRP, the overlying layers are indeed symplastically isolated through PDL5 action. This is clearly a central point in their model and something that the authors highlight in their abstract "PDL5, which functions to restrict the cell-to-cell movement of signals via plasmodesmata, is induced by auxin in cells overlying LRP in a progressive manner. PDL5 localizes to plasmodesmata and negatively impacts organ emergence as well as overall root branching. We present a model where auxin-directed spatiotemporal expression of PDL5 in LRP-overlying cells promotes the symplasmic isolation of those cells to help focus auxin distribution and modulate the timing of LRP emergence". As I already mentioned in my previous review, the authors do not present direct evidence to support this statement. The fact that PDL5 acts as a positive regulator of callose deposition in other tissues does not prove its function in LR primordia overlying tissues. In their response, the authors refer to Oparka's work (1995 J Exp Bot 46:187-197) and the use of CF dye as a better tool than SUC2::GFP to monitor symplastic connectivity during LR primordia development. However, in this paper there is no detailed analysis of symplastic connectivity around the developing LR primordia at early stages. Using a similar approach with CF or other phloem-mobile probes (Knoblauch et al. 2015 Plant Physiol 167 1211-1220) the authors have a great opportunity to experimentally prove that symplastic connectivity is indeed modified during LR development and compare wild-type and *pdl5* mutant. I fully agree with the authors that CF is a very good test indeed. This approach is straightforward, can rapidly implemented and given the exquisite cellular resolution the authors can achieve, will give a clear answer the impact of PDL5 on symplastic connectivity, and substantially add to the present paper.

Reviewer #2 (Remarks to the Author):

Unfortunately, I do not feel that the reviewers comments have been adequately addressed.

The request for data showing that the level of DR5 induced expression changes in overlying cells was met with the comment that the level of DR5:3venus is highly variable, and the variance in wildtype exceeds that of any difference between wildtype and the mutant or over-expressing lines. These data strongly argue against the presented model that PDL5 affects auxin accumulation in the overlying layers. At a minimum, the conflicting data need to be clearly included in the manuscript.

Making the experimental set up for the RT-PCR experiments match the stage at which other data are taken from is good science, and should be done. In figure 1(f), the request for more information identified that the control does not match the ecotype of the mutants. This should be corrected as well.

I made a mistake in the original review. The image in which PDL5 expression appears to be in the lateral primordium is figure 1f, not 1e as originally stated. I apologize for my mistake. The explanation given for the appearance of staining in the lateral root primordium (that it is bleed through from other layers) does not make sense for figure 1f.

Authors' Responses to Reviewers' Comments:

Reviewer #1 (Remarks to the Author):

In their response, the authors have only partially replied to my concerns. In particular, given the centrality of the reduction of symplastic movement in their model, I still think the authors should provide experimental data demonstrating, instead of postulating, that at early stages of LRP, the overlying layers are indeed symplastically isolated through PDL5 action. This is clearly a central point in their model and something that the authors highlight in their abstract “PDL5, which functions to restrict the cell-to-cell movement of signals via plasmodesmata, is induced by auxin in cells overlying LRP in a progressive manner. PDL5 localizes to plasmodesmata and negatively impacts organ emergence as well as overall root branching. We present a model where auxin-directed spatiotemporal expression of PDL5 in LRP-overlying cells promotes the symplasmic isolation of those cells to help focus auxin distribution and modulate the timing of LRP emergence”. As I already mentioned in my previous review, the authors do not present direct evidence to support this statement. The fact that PDL5 acts as a positive regulator of callose deposition in other tissues does not prove its function in LR primordia overlying tissues. In their response, the authors refer to Oparka's work (1995 J Exp Bot 46:187-197) and the use of CF dye as a better tool than SUC2::GFP to monitor symplastic connectivity during LR primordia development. However, in this paper there is no detailed analysis of symplastic connectivity around the developing LR primordia at early stages. Using a similar approach with CF or other phloem-mobile probes (Knoblauch et al. 2015 Plant Physiol 167 1211-1220) the authors have a great opportunity to experimentally prove that symplastic connectivity is indeed modified during LR development and compare wild-type and *pdlp5* mutant. I fully agree with the authors that CF is a very good test indeed. This approach is straightforward, can rapidly implemented and given the exquisite cellular resolution the authors can achieve, will give a clear answer the impact of PDL5 on symplastic connectivity, and substantially add to the present paper.

[Response: We have actioned the reviewer's helpful suggestion to implement a CF-based phloem loading approach. Accordingly, we have performed the phloem dye-loading assays on wild-type and *pdlp5-1* seedlings and examined unloading patterns of CF into un-emerged LRPs. The results are now included in the new Supplementary Figure 14, showing CF unloading into un-emerged LRP. It is noteworthy that, although CF unloading into late LRPs and emerged LRPs was easily discernible, it was not easy to identify nascent LRP having clear fluorescent signals. The CF signal was substantially lower in nascent LRPs compared to the CF signal in the vasculature—in the LR forming zone, the CF signal moved out of phloem into stele and pericycle. However, no CF diffusion was observed between the cells of the nascent LRP having CF signal and the cells overlying it. We interpret this data that it indicates that even the earliest stage LRPs are symplasmically isolated. This pattern of unloading was similar in *pdlp5-1* mutant seedlings. Thus, while it was a good experimental technique to try, unfortunately, this approach was not suitable to directly test the impact of PDL5 on cell-cell connectivity of overlying cells. However, although it is beyond the scope of our current manuscript, we thank the reviewer for the suggestion as the experiment revealed that nascent LRP is symplastically connected but symplasmically disconnected from its overlying cells. Recently, Berhin et al. (Cell, 176:1367-78, 2019) reported the finding that early LRPs form cuticle as a diffusion barrier, which may coincide

with the symplasmic disconnection of nascent LRP with overlying cells. We included two new paragraphs in the revised Discussion as follows, describing our attempts to test PDLP5's impact on cell-to-cell connectivity in overlying cells and the results and citing the reference regarding the formation of cuticle in early LRP.

“Due to the highly spatiotemporal nature of PDLP5 expression in the LRP-overlying cells, and our previous data proving that PDLP5 closes plasmodesmata, we hypothesized that the overlying cells might become symplasmically isolated during LRP emergence. However, it was reported by Benitez-Alonso et al. that GFP expressed under the phloem-specific SUC2 promoter was present in parental root cells including LRP-overlying endodermal cells during stages I-II¹³. Since PDLP5 is expressed in cells overlying nascent LRP, observing GFP within these cells would at first seem to contradict our hypothesis on PDLP5's role for plasmodesmal regulation in those cells. However, GFP can remain quite stable within tissue for many hours after expression. Thus, it is possible that the GFP had already moved from the phloem into outer cell layers prior to PDLP5 expression during early LRP formation, and the signal Benitez-Alonso et al. detected was actually residual fluorescence from GFP present in that tissue. Another technique for observing the potential effects of PDLP5 on cell-to-cell connectivity in LRP-overlying cells would, therefore, be to observe it in real-time as shown in the report by Oparka et al.²⁵. That report noted the unloading of phloem-loaded carboxy fluorescein (CF) dye into emerging LRP but no dye movement from cells within LRP to LRP-overlying cells. Our experiment showed that CF was unloaded into the cells of nascent (stage I-II) but no dye movement occurred from the cells of the LRP into cells overlying them, both in wild-type and the *pdlp5-1* roots (Supplementary figure 13). Our results seem to indicate that LRPs are not symplasmically connected with overlying cells, even from the earliest stages of development.

While these results meant we could not use CF phloem loading to directly test PDLP5-dependent isolation of LRP-overlying cells, it led to the unexpected insight that one of the earliest steps during LRP development is its isolation from outer root cells. This isolation may prevent the loss of important growth signals from the nascent LRP into the overlying tissues. Supporting this, it was recently discovered that a cuticle layer, functioning as a diffusion barrier, is deposited at the outermost cell wall of developing LRP at stage I-III²⁶; our results make it tempting to speculate that the symplasmic isolation of LRP from outer cells may coincide with the cuticle deposition. It would be an interesting future investigation to detail when and how the plasmodesmal disconnection occurs between LRP and overlying cells. Meanwhile, direct examination of the PDLP5's impact on cell-to-cell connectivity of LRP-overlying cells would have to wait until a new technique is developed, allowing real-time analysis of a movement tracer out of those cells.”

To reflect the reviewer's point, we have revised the statement in our abstract as follows.

“We present a model, incorporating the spatiotemporal expression of PDLP5 in LRP-overlying cells into known auxin-regulated LRP-overlying cell separation pathways, and speculate how PDLP5 may function to negatively regulate the lateral root emergence process.”]

Reviewer #2 (Remarks to the Author):

Unfortunately, I do not feel that the reviewers' comments have been adequately addressed.

The request for data showing that the level of DR5 induced expression changes in overlying cells was met with the comment that the level of DR5:3venus is highly variable, and the variance in wild type exceeds that of any difference between wild type and the mutant or over-expressing lines. These data strongly argue against the presented model that PDLP5 affects auxin accumulation in the overlying layers. At a minimum, the conflicting data need to be clearly included in the manuscript.

[Response: We apologize that our explanation in response was not satisfactory. We do agree with the reviewer that intensity measurement data should be included. The data is now included as a new Supplementary Figure 13 in our revised manuscript. We have also included the following text in the relevant Results section.

“The number of fluorescent overlying Co cells increased in *pdlp5-1* roots, while it decreased in *PDLP5OE* (Fig 4a; Supplementary Fig. 11b). We have also examined the DR5-3VENUS signal intensities in LRP-overlying Co cells, but the variance in the fluorescence intensity of DR5:3VENUS was too high to show any statistical differences (Supplementary Figure 13). The Box plot analysis of overlying Co cell numbers revealed that while 50% of WT seedlings had 3-5 DR5:3VENUS-positive Co cells, this range was skewed lower in *PDLP5OE*, with 50% of seedlings having only 2-4 fluorescent Co cells, and skewed higher in *pdlp5-1* mutants, with 50% having 4-5 positive Co cells (Fig. 4b). These results suggest that *PDLP5* negatively regulates the spread of auxin through overlying Co cells during LRP emergence.”]

Making the experimental set up for the RT-PCR experiments match the stage at which other data are taken from is good science, and should be done. In figure 1(f), the request for more information identified that the control does not match the ecotype of the mutants. This should be corrected as well.

[Response: We have now included the new RT-PCR data matching the stage at which other data are taken and replaced the old data in Supplementary Figure 3C. Here we included additional control for auxin response as well. The result reconfirms SA- and auxin-dependent PDLP5 induction.

We thank the reviewer for identifying what we had overlooked in our first submission. We had included the genetic background information in the figure legends during our first revision but to make it easily recognized in the figure, we have now included the information in the labels of Figure 1f as well.]

I made a mistake in the original review. The image in which PDLP5 expression appears to be in the lateral primordium is figure 1f, not 1e as originally stated. I apologize for my mistake. The explanation given for the appearance of staining in the lateral root primordium (that it is bleed through from other layers) does not make sense for figure 1f.

[Response: We also apologize for the confusion; bleed-through was inappropriately used in our previous response. What we meant in our previous response was that GUS staining sometimes creates diffusive patterns, either from overstaining or overclearing. Regardless, we do appreciate the reviewer’s point and have now included a set of images in new Figure 1f, showing that PDLP5 expression is in overlying cells in WT Col-0, not in LRP, the point which is also shown in Figure 1a. Also, we have resized the close-up LRPs in Supplementary Figure 1.

We hope that this helps to make it easier to see the difference of the GUS staining pattern of PDLP5pro:GUS from that of DR5:GUS.]

We thank both reviewers for their constructive and thoughtful comments. We hope that both reviewers find our responses and additional revisions made are now satisfactory.

REVIEWERS' COMMENTS:

Reviewer #1 (Remarks to the Author):

I acknowledge that the authors have made efforts to address some of my concerns related to symplastic movement. Although, the CF experiments did not confirm symplastic connection between nascent LRP and overlying cells at early stages, nor did they reveal any changes in cell-to-cell connectivity in PDLP5 overexpressing versus *pdlp5-1* known-out lines, the authors have now included these data in the revised manuscript and discuss their significance in the light of previous and recent work from Nawrath's lab. I greatly appreciate the frank attitude. Overall, the revised manuscript has merits added. As I said before, the work from Sager et al. represents an advance in the field of cell-to-cell communication and organ growth in plants and will benefit the field.

Minors points:

. Line 67-68: " In the current study, we report that our data suggest that ... " please rephrase.

. Line 132: "Following observing that PDLP5-GFP localizes to plasmodesmata in root cells treated with auxin (Fig. 1g)". I think the authors meant "Fig.1 e"

Reviewer #2 (Remarks to the Author):

Figure 1f is now satisfactory- provided the authors are confident that > 90 % of the LRP in their mutant look like this. If < 90% appear as shown, additional wording describing the variation is required. (I don't need to see the author's response to this, I just want to be clear as to what I require and how I am interpreting the authors response.)

The resizing of Supp Fig 1 closeups is appreciated and fine now.

The new CF data showing that LRP are isolated from the overlying cells argue against the arrow in the model (Fig 4 h) showing auxin moving from the LRP to the overlying cells. There could be specific transporters, but these have not been shown, so I suggest as a friendly recommendation that this arrow be removed along with the sentence in lines 306-308 describing the arrow. As stated (though not referenced) in line 302, auxin levels have been shown to rise in outer cell layers even before the founder cells (EMBO J (2013)32:149-158), so the induction of PDLP5 could occur cell autonomously in outer cell layers; there is no requirement for auxin to flow from the LRP to the endodermis. High levels of auxin in the endodermis could induce PDLP5 and thus restrict the number of cells undergoing a feed-forward increase in auxin response.

--

If the mutant fails to restrict the auxin-induced cell wall loosening associated with LR emergence, one wonders if the mutant might perhaps become more susceptible to attack by root pathogens (this is a friendly speculation, not one expected to impact the fate of this manuscript).

Authors' responses to REVIEWERS' COMMENTS:

Reviewer #1 (Remarks to the Author):

I acknowledge that the authors have made efforts to address some of my concerns related to symplastic movement. Although, the CF experiments did not confirm symplastic connection between nascent LRP and overlying cells at early stages, nor did they reveal any changes in cell-to-cell connectivity in PDLP5 overexpressing versus pdlp5-1 known-out lines, the authors have now included these data in the revised manuscript and discuss their significance in the light previous and recent work from Nawrath's lab. I greatly appreciate the frank attitude. Overall, the revised manuscript has merits added. As I said before, the work from Sager et al. represents an advance in the field of cell-to-cell communication and organ growth in plants and will benefit the field.

[Response: We appreciate the reviewer's acknowledgment about our effort in providing additional experimental data in response to his/her comments.]

Minors points:

. Line 67-68: " In the current study, we report that our data suggest that ... " please rephrase.

. Line 132: "Following observing that PDLP5-GFP localizes to plasmodesmata in root cells treated with auxin (Fig. 1g)". I think the authors meant "Fig.1 e"

[Response: We thank the reviewer for these minor comments. We made edits in the final revision.]

Reviewer #2 (Remarks to the Author):

Figure 1f is now satisfactory- provided the authors are confident that > 90 % of the LRP in their mutant look like this. If < 90% appear as shown, additional wording describing the variation is required. (I don't need to see the author's response to this, I just want to be clear as to what I require and how I am interpreting the authors response.)

[Response: We confirm that what we show in Figure 1f represent > 90 % of the LRP in the mutants we used.]

The resizing of Supp Fig 1 closeups is appreciated and fine now.

[Response: We appreciate the reviewer's remark.]

The new CF data showing that LRP are isolated from the overlying cells argue against the arrow

in the model (Fig 4 h) showing auxin moving from the LRP to the overlying cells. There could be specific transporters, but these have not been shown, so I suggest as a friendly recommendation that this arrow be removed along with the sentence in lines 306-308 describing the arrow. As stated (though not referenced) in line 302, auxin levels have been shown to rise in outer cell layers even before the founder cells (EMBO J (2013)32:149-158), so the induction of PDLP5 could occur cell autonomously in outer cell layers; there is no requirement for auxin to flow from the LRP to the endodermis. High levels of auxin in the endodermis could induce PDLP5 and thus restrict the number of cells undergoing a feed-forward increase in auxin response.

[Response: We greatly appreciate the reviewer's comment on this point. We agree with the reviewer's suggestion and removed the arrow from Figure 4h. To reflect his/her helpful suggestion, we have also added the following statements in the pertinent Discussion section.

“Alternatively, high levels of auxin accumulating in the En could be sufficient to induce PDLP5 and thus restricting the number of cells undergoing a feed-forward increase in auxin response. In this scenario, auxin-dependent PDLP5 expression in outer cell layers might occur cell-autonomously without needing auxin to flow from the LRP to the endodermis.”]

--

If the mutant fails to restrict the auxin-induced cell wall loosening associated with LR emergence, one wonders if the mutant might perhaps become more susceptible to attack by root pathogens (this is a friendly speculation, not one expected to impact the fate of this manuscript).

[Response: It is an interesting speculation. How the plant defense to root pathogens may be affected during the new organ emergence certainly deserved future investigations. We hope that the broad readership that Nature Communications has helps to disseminate our findings to many readers, stimulating new thoughts and ideas.]

We sincerely thank the reviewers for their constructive comments and enthusiasms about our manuscript.